**Investigation**

# Improved genotype inference reveals cis- and trans-driven variation in the loss-of-heterozygosity rates in yeast

Michael S. Overton (ID), Sergey Kryazhimskiy (ID)*

Department of Ecology, Behavior and Evolution, University of California San Diego, La Jolla, CA 92093, United States

*Corresponding author: Department of Ecology, Behavior and Evolution, University of California San Diego, La Jolla, CA 92093, United States. Email: skryazhi@ucsd.edu

Loss-of-heterozygosity (LOH) events are an important source of genetic variation in diploids and are implicated in cancer. LOH-event rates vary across the genome and across genetic backgrounds, but our understanding of this variation is incomplete. State-of-the-art measurements of LOH rates are obtained from mutation accumulation (MA) experiments in heterozygous hybrids, mainly in yeast *Saccharomyces cerevisiae*. These measurements hinge on the accuracy of inference of diploid genotypes from short sequencing reads. We analyzed a new large yeast MA dataset and found that the currently standard "single-reference" genotyping approach can lead to errors in LOH-rate estimates and produce spurious homolog biases. To address this problem, we developed a novel genotyping approach for MA experiments that is symmetric with respect to both homologs, removes dubious heterozygous markers, and corrects for undetected LOH events. We report revised estimates of LOH rates across 12 yeast hybrids, which differ by factors between 0.19 and 5.3 from previously published ones. Our revised estimates do not support the previously reported positive correlation between the rate of terminal LOH events and the hybrid heterozygosity. Finally, our analysis reveals that the 60-fold variation in the rates of interstitial LOH events across yeast hybrids is driven overwhelmingly by genetic factors with genome-wide (trans) effects. In contrast, the 6-fold variation in terminal LOH events is driven by both trans and local (cis) factors. Our results provide a foundation for reliable detection of LOH events and further investigations into the genetic underpinnings of LOH-rate variation.

Keywords: mutation accumulation; genotyping; terminal LOH; interstitial LOH; homolog bias

## Introduction

In diploid organisms, many genetic variants are heterozygous, that is, present only on one of the two homologous chromosomes. The phenotypic effects of such variants are often expressed only partially or not at all. If a heterozygous variant becomes homozygous, ie, present on both homologous chromosomes, its phenotypic effects are expressed fully, which can have profound consequences both for the individual (e.g. if the variant causes disease (Lapunzina and Monk 2011)) and for the population (e.g. if it provides an adaptive advantage (James et al. 2019)). Homozygotization can occur either through sexual reproduction or through so-called loss-of-heterozygosity (LOH) events that result from mitotic recombination during vegetative growth (Heil 2023). As with some other types of mutations, LOH events occur as a byproduct of processes that repair DNA damage (Symington et al. 2014). LOH events are among the most common types of mutations and each event can affect hundreds of kilobases (Sui et al. 2020). Consequently, they play an important role in the evolution of populations of organisms (Heil 2023) as well as populations of cancer cells within organisms (Negrini et al. 2010). However, despite their importance, estimates of LOH rates became available only recently (St Charles et al. 2012; Dutta et al. 2017, 2021; James et al. 2019; Tattini et al. 2019; Loeillet et al. 2020; Pankajam et al. 2020; Sui et al. 2020; Tutaj et al. 2022), and our understanding of the variation in these rates across the genome and across organisms is far from complete.

Recent work has demonstrated that LOH events are important in a range of biological contexts. They are a major driver of genetic diversity in a variety of organisms (Peter et al. 2018; Dale et al. 2019; Bulankova et al. 2021; Cruz-Saavedra et al. 2022). They can promote adaptation by exposing recessive beneficial alleles to selection (Mandegar and Otto 2007; Gerstein et al. 2014; Heil et al. 2017; Marad et al. 2018; Dale et al. 2019; James et al. 2019), reduce deleterious load (Omilian et al. 2006; Heil 2023), facilitate fitness valley crossing (Schoustra et al. 2007; Baselga-Cervera et al. 2022), and alleviate negative epistasis (Anderson et al. 2010; Li et al. 2013; Lancaster et al. 2019; Bozdag et al. 2021). The latter can help resolve hybrid incompatibilities and thereby promote both intra- and interspecific hybridization (Heil et al. 2017). In the disease context, LOH events have been implicated in the development of several types of cancers (Spandidos et al. 1997; Lindblad-Toh et al. 2000; McKinnon and Caldecott 2007; Lapunzina and Monk 2011; Alves et al. 2017). For example, LOH events can lead to the homozygotization of loss-of-function mutations in the gene MLH1/MSH2, which promotes colorectal cancer (Zhang et al. 2006), and in the gene ARH1, which promotes ovarian cancer (Feng et al. 2008).

The importance of LOH events, particularly for cancer development, has been recognized for decades (Naylor et al. 1987; Menon et al. 1990), and early genetic studies have dissected various mechanisms of mitotic recombination that produce LOH events (reviewed by Pâques and Haber (1999)). More recently, researchers began characterizing LOH events at the genome scale (Lee et al. 2009; St Charles et al. 2012; Yim et al. 2014; Dutta et al. 2017, 2021; James et al. 2019; Tattini et al. 2019; Loeillet et al. 2020; Pankajam et al. 2020; Sui et al. 2020; Tutaj et al. 2022; Vijayan et al. 2025). Much of our understanding of LOH rates and

distributions is based on work carried out with yeast *Saccharomyces cerevisiae*. These studies have identified two types of LOH events, which are driven by different molecular mechanisms and have distinct properties (Symington et al. 2014; Jinks-Robertson and Petes 2021). The more frequent interstitial LOH (iLOH) events are localized within a chromosome and typically cover hundreds to thousands of base pairs. The less frequent terminal LOH (tLOH) events extend to one end of a chromosome and typically encompass thousands to hundreds of thousands of base pairs. Because tLOH events are so large, they are thought to be the main drivers of the loss of heterozygosity in the yeast genome (Sui et al. 2020; Dutta et al. 2021). Several studies reported that LOH event rates vary across the genome, with tLOH events being enriched near telomeres (Sui et al. 2020) and at certain localized hotspots, most notably on the right arm of chromosome XII (Song et al. 2014; Zheng et al. 2016; Peter et al. 2018; James et al. 2019; Pankajam et al. 2020; Sampaio et al. 2020; Sui et al. 2020; Zhang et al. 2022). In one of the most comprehensive studies to date, Dutta et al estimated how the rate of LOH events as well as the probability that a typical site in the genome is covered by an LOH event (which we refer to as the "LOH conversion rate") vary across 9 diverse, intraspecific hybrids of *S. cerevisiae* and found large variation in these rates (Dutta et al. 2021). They also reported a positive correlation between the hybrid's heterozygosity and its rate of tLOH events, which is unexpected given that the DNA repair complex is thought to favor higher homology (Datta et al. 1997). Contrary to this observation, two other studies found that LOH event rates are lower in more heterozygous hybrids (Tattini et al. 2019; Tutaj et al. 2022). The causes of this discrepancy are unclear, and how the degree of divergence between homologous chromosomes affects the rate of LOH events (if at all) remains unresolved.

Overall, previous studies provide an initial picture of variation in LOH events in the *S. cerevisiae* clade. However, they also raise several questions of both methodological and biological nature. Methodologically, most of the recent genome-wide LOH studies are based on mutation accumulation (MA) experiments in hybrids that are heterozygous at $\sim 10^4$ to $10^5$ sites distributed across the genome (St Charles et al. 2012; Dutta et al. 2017, 2021; James et al. 2019; Tattini et al. 2019; Loeillet et al. 2020; Pankajam et al. 2020; Sui et al. 2020). While this approach is powerful, analyses of the resulting sequencing data face two potentially serious challenges. First, an accurate identification of LOH events relies critically on the ability to correctly genotype ancestral and descendant clones. Most current genotyping methods rely on mapping of short-read Illumina sequences onto a single reference genome (Lee et al. 2009; St Charles et al. 2012; Yim et al. 2014; Laureau et al. 2016; Dutta et al. 2017, 2021; James et al. 2019; Tattini et al. 2019; Loeillet et al. 2020; Pankajam et al. 2020; Sui et al. 2020; Mozzachiodi et al. 2021; Vijayan et al. 2025). However, such genotyping is biased (Minoche et al. 2011; Bobo et al. 2016) and can potentially cause errors in LOH counts as well as lead to apparent homolog biases among detected LOH events (Loeillet et al. 2020; Pankajam et al. 2020). The most common strategy for mitigating these errors is to only consider LOH events that are supported by conversions of multiple adjacent markers (Dutta et al. 2017, 2021; Loeillet et al. 2020; Pankajam et al. 2020; Sui et al. 2020). But this approach is also problematic because it removes an unknown number of true (mostly short) LOH events, leading to biases in the rates and lengths of LOH events.

The second methodological challenge is that not all LOH events are equally detectable. The probability of detecting an LOH event depends on its length (shorter events are harder to detect) and on the number of heterozygous markers that it can affect (fewer markers reduce the detection probability). Furthermore, marker-based approaches can detect at most one tLOH event on a given chromosome arm in a given lineage. Correcting for the detection bias is especially critical when comparing LOH rates across hybrids with different numbers of heterozygous markers, but previous studies failed to do so. These methodological problems raise some questions about the reliability of current LOH estimates.

One of the most biologically interesting unresolved questions concerns the causes of variation in the rates of LOH events across the genome and across genetic backgrounds (Dutta et al. 2021). One aforementioned hypothesis is that LOH rates are determined by the ability of the DNA-repair machinery to recognize homology (Tattini et al. 2019; Dutta et al. 2021; Tutaj et al. 2022). Another complementary hypothesis is that local LOH rates are affected by various DNA elements, such as large repeats (e.g. the yeast rRNA gene array), histone markers, GC content, etc (Zheng et al. 2016; Sui et al. 2020). More generally, we can categorize genetic factors that affect LOH events into those that have genome-wide effects ("trans" factors) and those that have local effects in the genomic neighborhood of where these factors are themselves located ("cis" factors). While previous studies identified certain specific factors that affect LOH event rates, no attempt has been made to systematically quantify the contribution of various (possibly unknown) cis and trans factors to the variation in iLOH and tLOH rates.

In this paper, we first address the methodological issues described above. To this end, we develop an improved genotyping pipeline for analyzing MA experiments in hybrids. The key idea behind our approach is to map the sequencing reads from each clone onto both parental reference genomes and then filter out unreliable genotype calls in a statistically sound way. We also develop corrections for undetected LOH events. We then apply our improved genotyping pipeline to existing MA datasets in yeast and provide more reliable estimates of LOH rates and characteristics. Finally, we use these revised estimates to interrogate genetic factors that drive the variation in the LOH rates in the *S. cerevisiae* clade.

## Methods
### Experimental work
*Strains and media*

Strains and media are described in Overton et al. (2023) and are listed in Supplementary Table 7 in Supplementary File 2. Briefly, we generated 3 types of founder strains of *Saccharomyces cerevisiae* by mating haploid derivatives of strains BY4741 (Brachmann et al. 1998) (BY) and YAN501 (Brem et al. 2002; Ba et al. 2022) (RM), which vary at approximately 40,000 SNPs (Bloom et al. 2013). Founder strains are thus heterozygous at 40,000 sites, and differ only at the ADE2 locus, which was disrupted by the homozygous insertion of 1 of 3 genetic cassettes. All these cassettes carry a resistance marker, one also carries a Cas9 gene, and another one carries a gene drive element (Cas9 gene + guide RNA gene). Since the presence of the latter two cassettes was found to have negligible effects on LOH characteristics (see Overton et al. 2023 for details), in the present work, we treat all founders and their descendants as replicates. Sixteen founder clones were isolated (we will refer to them as "founders" or "ancestors" interchangeably) and each of them was used to establish between 10 and 12 replicate mutation accumulation (MA) lines, for a total of 285 MA lines. We refer to all the lines descendent from a given founder as a "family." We propagated all MA lines for 800 generations on YPD

agar by bottlenecking random colonies to single cells after 48 h of growth at 30 °C.

After sequencing (described below), we found that library preparation had failed for 5 founders (F_D00, H_F00, H_H00, N_F00, N_H00), thus we excluded all 64 descendant end-point clones in these families from further analyses. Two additional MA families (F_B and F_E; 2 founders and 27 end-point clones) were apparently triploid and were also excluded. Finally, 19 end-point clones were omitted due to failed sequencing quality control. In total, we removed 110 end-point clones and retained a final set of 175 end-point clones (Supplementary Table 6 in Supplementary File 2).

### Illumina sequencing

We sequenced all founder and end-point clones to a median depth of 36.7× using 100 bp paired-end Illumina Nextera library prep and Hi-Seq 4000 platform (Baym et al. 2015).

### Genotype confirmation

We selected 3 genomic regions to verify the genotypes called by our method. Region 1 spans positions 956964–957730 (all positions are given with respect to the S288C reference genome R64) on the right arm of Chr XII and contains 3 markers. Region 2 spanned positions 1030968–1031837 on the right arm of Chr XV and contains 6 markers. These "control" regions were selected due to their relatively high conversion rates, but even homolog bias. Region 3 spans positions 103380–103869 on the left arm of Chr XII and encompasses 5 HCHB markers (amplification primers are provided in Supplementary Table 8 in Supplementary File 2). We selected 2 ancestors and 7 end-point clones (Supplementary Table 2 in Supplementary File 2) from our MA experiment, PCR amplified these 3 regions, and Sanger sequenced them.

## Data

We searched the literature for studies that measured LOH rates using highly heterozygous hybrid yeast strains under mutation accumulation conditions. To directly compare the sets of LOH events detected with our revised pipeline against previously established methods, we required studies to provide both raw sequencing data and the identities and boundaries of all detected LOH events. We excluded studies using mutated strains (e.g. knockouts of repair genes) and those that artificially induced LOH events (e.g. via the HO locus). We found 3 studies that complied with these requirements (Pankajam et al. 2020; Sui et al. 2020; Dutta et al. 2021). One study (Sui et al. 2020) examined LOH events in 93 MA lines derived from a single hybrid strain, 10 of which were propagated for 1,500 generations and 83 were propagated for 3,000 generations. A second study (Pankajam et al. 2020) investigated 2 hybrid strains, with 10 MA lines established from each and propagated for 2,000 generations. The third study (Dutta et al. 2021) investigated 9 hybrid strains, each of which was used to establish between 12 and 20 MA lines that were propagated for between 1,772 and 2,452 generations. We downloaded sequencing reads for these clones from the SRA database of NCBI. Data downloads and/or processing steps failed for some number of clones, and we also excluded 13 clones from the Dutta et al. study that had become homozygous across over 90% of their genomes. Altogether, we analyzed data from 424 end-point clones across 12 hybrid yeast strains, which are listed in Supplementary Tables 7 and 9 in Supplementary File 2.

We performed genotyping with our reference-symmetric pipeline and detected LOH events as described below. We obtained the reported LOH events from the respective journal websites and processed them into a common format with custom R scripts.

Note that there are important differences in the definitions of LOH events across the 4 studies (including our own). First, the definition of what constitutes an LOH event is not the same across studies. For example, we only required 1 or more converted markers in a clone to include it as an LOH event, while Pankajam and Dutta required at least 2, and Sui at least 3 markers. Complex LOH events, those that include runs of converted markers with unconverted markers or those converted to the other homolog, were also treated differently across studies. Pankajam counted each LOH tract, or contiguous run of homozygous markers, as an independent event. Sui merged adjacent LOH tracts into single, complex events if they were separated by less than 10 kb. Dutta merged adjacent LOH tracts if separated by 2 or fewer unconverted markers. Unless otherwise specified, we report the unmodified set of events provided by each study.

## Genotyping
### Single-reference genotyping

If the raw data contained untrimmed reads (as assessed by fastQC (Andrews 2010), default settings), reads were trimmed using Trimmomatic (Bolger et al. 2014) v0.39 in paired-end mode with the following settings: LEADING:10, TRAILING:10, CROP:200, SLIDINGWINDOW:6:10, MINLEN:30, and the appropriate adapter sequence file. We then mapped the reads with bwa-mem2 (Li and Durbin 2009; Li 2013) against the parental reference genome, with the default settings. As the reference genome for single-reference genotyping, we followed previous studies (Dutta et al. 2017, 2021; Liu and Zhang 2019; Pankajam et al. 2020) and used the *S. cerevisiae* reference genome R64 (NCBI RefSeq GCA_000146045.2) derived from the S288C strain. For dual-reference genotyping, we assembled alternative reference genomes (see Section "Dual-reference genotyping").

We performed initial genotyping with GATK (McKenna et al. 2010; van der Auwera and O'Connor 2020). This consisted of inferring preliminary genotypes with HaplotypeCaller (settings: -ERC) on a per clone basis, then grouping clones into families (all end-point clones descendant from a MA founder, and the founder, when applicable) and performing a second genotyping step with GenotypeGVCFs (Poplin et al. 2018) (default settings). This second, joint, calling step is meant to improve final genotype inferences in each individual sample by integrating mapping, genotyping, and pedigree information across samples (DePristo et al. 2011; Poplin et al. 2018).

All indel variants were excluded, while putative SNP variants were filtered out under any of the following conditions: QD < 2, MQ < 20, FS > 60, or SOR > 10 (DePristo et al. 2011; Poplin et al. 2018). These filters remove low-quality calls conditioned on read depth, poor mapping quality, and high bias in read mapping to each DNA strand, and high bias in allele-specific read mapping to each strand, respectively. All SNP variants were required to be biallelic, and supported by at least 6 reads and at most 5 times the sample-specific median depth (because regions with unusually high coverage probably represent copy-number variants and are difficult to map correctly); otherwise, they were removed. Ancestral heterozygous variants are expected to be present across many, if not all, clones. Thus, calls with a site-wise QUAL score < 1,000 (poor support for the presence of a variant site) were removed. We excluded calls at repeat regions, as annotated by RepeatMasker (Smit et al. 2015) on the S288C reference with default settings, and telomeric regions within 7.5 kb of chromosome arm ends, as they tend to compromise genotyping inferences from short read data (Cechova 2021; Ba et al. 2022). These procedures excluded 14.7% (1,774,809 bp) of the genome from the detection

of heterozygous markers. Therefore, when calculating per base-pair rates, we use the effective genome length of 12,071,326−1,774,809 = 10,296,517 bp.

For each variant, the genotyping algorithm outputs a confidence score (GQ) defined as the $\log_{10}$ likelihood ratio between the most and second most likely genotypes (McKenna et al. 2010; Poplin et al. 2018). The GQ score indicates the degree to which the read and alignment data support one genotype over any of the others. We therefore exclude all variants with GQ < 50.

Since genotyping is carried out on a family basis (i.e. jointly among clones that share a common ancestor) rather than on a clone-by-clone basis, we obtain a separate call set for each family, such that each site in this call set is polymorphic (biallelic) within the family.

### Aneuploidy detection and exclusion

We detect aneuploidy events by calculating the mean number of reads supporting each parental allele for each chromosome in each clone and normalizing these values by the mean genome-wide depth of that clone. Chromosomes for which the relative mean coverage of an allele was near zero (<0.05) or greater than 1.5 were marked as aneuploids. All aneuploid chromosomes were excluded from all LOH analyses.

We detect whole-genome aneuploidies by calculating the ratio of the mean coverages of the 2 parental alleles for each chromosome. We expect this ratio to be near 1 across all chromosomes for diploids, while a ratio of 0.5 or 2 in all chromosomes suggests that one set of parental chromosomes is present at twice the frequency of the other parental set. Two entire families (F_B and F_E) had ratios near 0.5, indicating whole RM-genome duplication, all of which were removed from LOH analyses.

### Dual-reference genotyping

*Construction of parental reference genomes.* The first step of our revised genotyping pipeline is to construct the parental reference genomes, i.e., modify the annotated fully assembled S288C reference genome to incorporate the differences between the parent and S288C. To this end, for BY and RM strains, we downloaded large contigs derived from long read sequencing from NCBI under accession numbers ASM2970261v1 (BY) and ASM2970263v1 (RM). For the other strains, parental sequencing data were only available in short-read (100–150 bp paired-end) form (Supplementary Table 10 in Supplementary File 2). Then, we performed mapping and genotyping on contig-level data with the Nucmer and delta2VCF modules from the Mummer4 program (Marçais et al. 2018) (default settings). For short read data, we performed mapping and genotyping with the single reference pipeline described above (acquiring the variants directly from HaplotypeCaller). Since the sequencing data for each hybrid parent were generated from haploid clones, we expect that 100% of reads from sites that differ between the parent and S288C reference should support the parental allele. Therefore, we filtered variants to only biallelic homozygous alternative genotypes (GT = 1/1). We then reconstructed the parental genome from the S288C reference and this set of variants with bcftools consensus (Li 2011) (settings: -H A). This program also outputs a chain file that lists the coordinates of each genomic position within the parental and the S288C sequences. This chain file is used to convert (liftover) the positions of variants generated from each parental reference to the common S288C position indices using a custom R script using the method described in (Hinrichs et al. 2006).

*Reference-symmetric read mapping and genotype calling.* For each evolved and ancestral (where applicable) clone, we apply the single-reference pipeline described above with respect to each parental reference genome, except we skip the last step, i.e., we keep all variants, including those with GQ < 50. This step results in two genotype call sets, one with respect to each parental reference. Then we apply the following series of conditions at each marker site to reconcile possible disagreements between the two calls and arrive at the final genotype call:

1) If both calls have GQ < 50 or if at least one call is missing, the site is discarded;
2) If the calls agree, this call is final;
3) If the calls disagree and
   a) the difference in the GQ scores is ≥ 30, the higher-quality call is final;
   b) the difference in the GQ scores is < 30, the site is discarded.

As with single-reference genotyping, the final call set that we obtain here is family-specific.

### Estimation of genotype call error rates

The fact that our dataset (Overton et al. 2023) contains sequence information for both the founder and endpoint clones for each MA line allows us to estimate the rate of Type I and Type II errors for our genotype calls. The basic idea is that the genomes of all end-point clones within the same family must be identical to the genome of their founder, except for those sites where either a new mutation or an LOH event occurred during the MA experiment. Since LOH and mutations are relatively rare events, it is unlikely that many independent MA lines share the same mutation or LOH event. Thus, a genotyping error in the founder can be identified as a mismatch between the genotype call in the founder and the plurality of genotype calls in its end-point clones.

Since the true genotypes are unknown, we develop a model for estimating the posterior odds ratio $R_{\text{false hom}}$ that the ancestor is a false homozygote at a given site (i.e. it is a heterozygote erroneously genotyped as a homozygote) and the posterior odds ratio $R_{\text{false het}}$ that the ancestor is a false heterozygote at a site (i.e. it is a homozygote erroneously genotyped as a heterozygote). Our model is described in detail in the Supplementary Text 1. It depends on 5 probabilities: the probability $\mu$ that a new mutation occurs at a given site in a given MA line during the MA experiment, the probability $\lambda$ that an LOH event converts this site to a homozygous state, the probability $f$ that the founder is homozygous at a polymorphic site, and the false homozygote and false heterozygote error probabilities $\epsilon_{\text{HOM}}$ and $\epsilon_{\text{HET}}$. More precisely, $\epsilon_{\text{HOM}}$ is the probability that a site that is in fact heterozygous is genotyped as homozygous, and $\epsilon_{\text{HET}}$ is the probability that a site that is in fact homozygous is genotyped as heterozygous. Note that in the analysis of sequencing data, we discard sites where more than 2 alleles are detected. Therefore, our model also considers only those mutations that flip the genotype of the MA line at a marker site between the heterozygous and homozygous states, which constitute about one-third of all mutations at any given marker site, and we disregard the remaining two-thirds of mutations that generate new alleles. In this model, we find

$$R_{\text{false hom}} = \frac{\epsilon_{\text{HOM}}}{1-\epsilon_{\text{HET}}} \frac{1-f}{f} \left(\frac{p_{10}}{p_{00}}\right)^{K_0} \left(\frac{p_{11}}{p_{01}}\right)^{K_1} \left(\frac{p_{12}}{p_{02}}\right)^{K_2} \tag{1}$$

$$R_{\text{false het}} = \frac{\epsilon_{\text{HET}}}{1-2\epsilon_{\text{HOM}}} \frac{f}{1-f} \left(\frac{p_{00}}{p_{10}}\right)^{K_0} \left(\frac{p_{01}}{p_{11}}\right)^{K_1} \left(\frac{p_{02}}{p_{12}}\right)^{K_2} \tag{2}$$

where $K_0$, $K_1$, and $K_2$ are the numbers of end-point clones genotyped as homozygous for the ancestral allele, heterozygous, and homozygous for the derived allele the focal marker site, respectively, such that $K_0 + K_1 + K_2 = n$, with $n$ being the total number of end-point clones that are descendant from the given founder and are genotyped at the focal site. $p_{ij}$ is the probability for observing a descendant clone with $j$ derived alleles given that the ancestor had $i$ derived alleles (see Supplementary Text 1 for the precise definitions of $p_{ij}$ and the derivation of expressions (1), (2)).

*A priori estimates of parameters.* Zhu et al. estimated the mutation rate in diploid yeast to be $1.67 \times 10^{-10}$ per bp per generation (Zhu et al. 2014). Therefore, for our 800-generation-long MA experiment, we set $\mu = 4.45 \times 10^{-8}$ (since only about a third of mutations flip between 2 existing alleles). Sui et al. estimated that the LOH conversion rate varies between $3.8 \times 10^{-6}$ and $1.6 \times 10^{-4}$ per bp per generation in yeast, depending on the chromosomal position (Sui et al. 2020). Thus, we conservatively set $\lambda = 0.128$. Finally, the probability $f$ that a polymorphic site in the ancestor is homozygous can be constrained by the fraction $f_{OBS}$ of homozygous sites observed in the ancestral call set because

$$f_{OBS} = f(1 - \varepsilon_{HET}) + (1 - f)\varepsilon_{HOM} \tag{3}$$

Thus, we have $f = (f_{OBS} - \epsilon_{HOM})/(1 - \epsilon_{HOM} - \epsilon_{HET})$. Equation (3) also constrains the values of $\epsilon_{HOM}$ and $\epsilon_{HET}$. Since $f_{OBS}$ is a convex combination of $1 - \epsilon_{HET}$ and $\epsilon_{HOM}$ and assuming that $\epsilon_{HOM} \ll 1$, the values of $f_{OBS}$ are confined to the interval $[\epsilon_{HOM}, 1 - \epsilon_{HET}]$, which implies that $\epsilon_{HOM} \leq f_{OBS}$ and $\epsilon_{HET} \leq 1 - f_{OBS}$.

*Homo-FDR.* We evaluated how the false-homozygote odds ratio, given by equation (1), with $f$ defined by equation (3), depends on the error probabilities $\epsilon_{HOM}$ and $\epsilon_{HET}$ for $f_{OBS} = 0.115$, which corresponds to the single-reference genotyping procedure with S288C reference genome, and for $f_{OBS} = 0.0154$, which corresponds to our final dual-reference genotyping procedure. We probed the range of $\epsilon_{HOM}$ and $\epsilon_{HET}$ between $10^{-4}$ and 0.1. We found that if at least 6 end-point clones are genotyped at a site where the founder is called homozygous and if all of them are called heterozygous, i.e. if the configuration of end-point clones is maximally discordant with the ancestor, then the odds ratio favors (for most parameters, overwhelmingly) the hypothesis that the site is in fact heterozygous in the founder (i.e. the homozygous call in the ancestor is an error; Supplementary Fig. 2 in Supplementary File 1). Therefore, we label such sites as false homozygous in our data (Supplementary Table 1 in Supplementary File 2). We then estimate the false homozygous discovery rate (Homo-FDR) as the ratio of false homozygous sites to the total number of sites genotyped in the founder, where genotype calls are available for at least 6 descendant end-point clones.

*Hetero-FDR.* We similarly evaluated the false-heterozygote odds ratio, given by equation (2), and also found that, if at least 6 end-point clones are genotyped at a site where the founder is called heterozygous and if all of them are called homozygous for the same allele (i.e. again, forming a configuration maximally discordant with respect to the ancestor), then the odds ratio favors the hypothesis that the site is in fact homozygous in the founder (i.e. the heterozygous call is erroneous; Supplementary Fig. 2 in Supplementary File 1). Therefore, we label such sites as false heterozygous in our data (Supplementary Table 1 in Supplementary File 2). We then estimate the false heterozygous discovery rate (Hetero-FDR) as the ratio of false heterozygous sites to the total

number of sites genotyped in the founder, where genotype calls are available for at least 6 descendant end-point clones.

## Heterozygous genotype filter

Under the simplest model of the genotyping process, the distribution of reads supporting either allele at a heterozygous site should follow the binomial distribution with success probability 0.5. However, this model does not capture certain errors and biases that may arise during PCR amplification and read mapping (Schwartz et al. 2011; Bobo et al. 2016; Hwang et al. 2019). Thus, the read distribution may deviate from binomial, e.g., it may have a different mean and higher variance. To this end, we model the read count supporting the RM allele as a beta-binomial random variable with the number of trials $n$ equal to the total read depth. This model captures a binomial random process where the success probability is itself drawn randomly from the beta distribution with parameters $\alpha$ and $\beta$; thus, it can account for potential biases as well as overdispersion relative to the binomial distribution. To fit the parameters of this model, we identify a set of 29,870 "high-confidence" ancestral heterozygous sites from our dual-reference call set as those called heterozygous in at least 15 out of 16 founders and in at least two-thirds (117/175) of end-point clones. We bin these calls across all founder clones by total read depth. For each read depth, we calculate the mean and the variance in the RM allele count. Read-depth bins with fewer than 100 calls are discarded due to high error in our variance estimates. We find that the mean is very close to $0.5n$, indicating the absence of systematic biases at high-confidence heterozygous sites (Supplementary Fig. 4a in Supplementary File 1). Therefore, we set $\alpha = \beta$, such that the variance in the read count is given by $\sigma^2 = n/4 + \rho n(n-1)$, where $\rho = (8\beta + 4)^{-1}$ captures the deviation from the binomial distribution. We find that our data are described very well by this model with $\rho = 2.09 \times 10^{-3}$ ($\beta = 59$; Fig. 3a), which explains 90.1% of the variation in read count variance. These results are robust with respect to changes in the thresholds that we use to define high-confidence heterozygous sites (Supplementary Fig. 5 in Supplementary File 1). These results suggest that the binomial model adequately captures the genotyping noise at low read counts ($n \ll 120$) but not at higher read counts.

We discard all heterozygous calls for which the read count supporting the RM allele falls outside of the 95% C.I.s of the beta-binomial distribution with $\alpha = \beta = 59$. This resulted in the removal of 220,618 out of 6,386,206 (3.5%) heterozygous calls from our dataset. In principle, we could reassign genotypes falling outside of our C.I.s as homozygous, analogous to the allele frequency-based genotyping employed in previous studies (Laureau et al. 2016; Tattini et al. 2019). However, this approach ignores all the other information integrated by the calling algorithm (e.g. base call quality, site-specific statistics) that caused it to emit a heterozygous call. The very fact that one piece of information, alternative read frequency, contradicts these other pieces of information is exactly why these calls should be excluded, since the data does not provide strong support for any call.

## Detection of LOH events
### Identification of ancestral markers

As discussed below in the main text, the accuracy of detection of LOH events hinges on the fact that the ancestral markers—i.e. sites heterozygous in the MA ancestor(s)—are identified accurately. To identify high-confidence ancestral markers, we first define the set of "potential markers", i.e. those sites where the parental

reference genomes differ from each other. In principle, ancestral hybrids should be heterozygous at all such sites. In practice, not all potential markers may actually be heterozygous in the ancestors, e.g., because ancestors may differ slightly from the sequenced reference strains or because LOH events occurred between mating and the beginning of the MA experiment. Thus, we reduce this set by incorporating information from sequencing data obtained from end-point clones as well as diploid ancestors when available.

### The set of potential markers

To obtain the set of marker sites that can potentially be heterozygous in the diploid hybrid ancestors of the MA experiment, we take the union of the 2 sets of variants where each parent differs from the S288C reference genome (see Section "Dual-reference genotyping") and remove their intersection. This effectively merges all of the variant positions found on the 2 parental genomes, and ignores positions where both parents differ from S288C. This procedure results in a set of all potential marker sites, i.e., positions where the ancestral diploid strains can potentially be heterozygous.

*Final set of ancestral markers.* We obtain the final set of high-confidence ancestrally heterozygous markers as follows. If the whole-genome sequencing data for the ancestral strain is available, then we retain only those potential markers that are called heterozygous (ii) in the ancestor and (ii) in at least 3 end-point clones descended from the ancestor. If the whole-genome sequencing data for the ancestor is not available, then we retain only those potential markers that are called heterozygous in at least 4 descendant end-point clones. For each ancestral hybrid, this procedure produces the final set of high-confidence markers that are heterozygous in that ancestor. From this point on, we refer to these sites simply as "markers" or "marker sites."

### Identification of LOH events

In each end-point clone, we identify "LOH tracts" as sequences of adjacent markers homozygous for the same allele ("converted markers"). In contrast to previous studies, we include LOH tracts supported by a single converted marker because short (<200 bp) LOH events may be common (Palmer et al. 2003). If an LOH tract contains the marker closest to the end of a chromosome, we assume that this tract extends to the end of the chromosome. Internal boundaries of an LOH tract are defined as the midpoint between the first or last converted marker and the adjacent unconverted marker. We define an LOH event as terminal (tLOH) if one of its boundaries is the end of a chromosome; otherwise, an LOH event is defined as interstitial (iLOH).

### Homolog assignment and the calculation of homolog bias

DNA repair processes can be complex, involving dissolution and re-assembly of various repair components, which can result in LOH events that contain a series of homozygous and heterozygous tracts (Hum and Jinks-Robertson 2017). Thus, following Sui et al. (2020), we merge LOH tracts into a single LOH event if their boundaries are separated by less than 10 kb. If an LOH event is composed of tracts converted to different homologs, we ascribe to it the homolog with the greatest total conversion length. We calculate the homolog bias in LOH events for a given hybrid as the number of events favoring each parental homolog divided by the total number of events. We then perform the same calculation for interstitial and terminal events independently.

### Corrections for undetected LOH events

To be detected in our analysis, an LOH event must convert at least one marker site. There are 3 possibilities for an LOH event to remain undetected. First, an LOH event may not affect any markers, i.e. it may fall between them. Second, if 2 overlapping LOH events occur in the same MA line, they appear as one. Third, since our genotyping procedure excludes telomeric regions (see Section "Single-reference genotyping" above), we cannot detect tLOH events shorter than 7.5 kb. We use the procedures described below to correct for the first 2 types of undercounts. We account for the third type of undercount by adjusting the denominator in the calculation of iLOH rates.

*Inter-marker LOH events.* Shorter LOH events have a higher probability of falling between markers and remaining undetected. We can estimate the probability of detecting an LOH event of length $\ell$ under the simplifying assumption that both markers and LOH events are uniformly distributed across the genome. In this model, an LOH event of length $ll\ell$ remains undetected if none of the $n$ markers occur inside it, which happens with probability $(1 - \ell/L)^n$. Thus, the probability of detection is $P_{\text{detect}}(\ell) = 1 - (1 - \ell/L)^n$. If $k_\ell$ events of length $\ell$ are actually detected, the expected number of LOH events of this length that have actually occurred is $k_\ell/P_{\text{detect}}(\ell)$. Since $P_{\text{detect}}(\ell) \approx 1 - e^{-n\ell/L}$, which approaches 1 for events with length $\ell \gg L/n$, i.e. those whose length substantially exceeds the average inter-marker distance.

To correct for undetected inter-marker LOH events, we bin the detected events by their $\log_{10}$-transformed lengths (since the distribution of lengths typically spans multiple orders of magnitude). We estimate the total number of events in each bin by applying the above formula with $\ell$ being the mean length of detected events in that bin. For bins that are empty but flanked on both sides by nonempty bins, we estimate the number of events in the empty bin by using a pseudocount 1. Clearly, since we have no information about events of very short length, this procedure does not fully correct for all missed events, nor does it fully eliminate the bias against short events in our length distribution.

*Overlapping LOH events.* We can roughly estimate the probability that 2 iLOH events occurring in the same clone overlap as follows. Suppose that $k$ iLOH events occur in the same clone, each with the typical length $\ell$, and they are distributed uniformly across the genome of length $L$. Then, the probability that event $i$ overlaps with event $j$ is $2\ell/L$, and the probability that at least one pair of $k$ events overlap is $P_{\text{overlap}} = 1 - (1 - 2\ell/L)^{k(k-1)/2}$. Given that the yeast genome size is $L = 12.07$ Mb, and given that we typically observe $k = 5.45$ iLOH events per clone with a median length of $\ell = 2.27$ kb, we estimate $P_{\text{overlap}} = 0.46\%$. Thus, we ignore these overlaps.

We can similarly estimate the probability that an iLOH event overlaps with a tLOH event. The probability that a particular iLOH event with length $\ell$ overlaps with a particular tLOH event with length $s$ is $(s + \ell/2)/L$, and the probability that at least one out of $k$ iLOH events overlaps with at least one out of $m$ tLOH events is $P_{\text{overlap}} = 1 - (1 - (s + \ell/2)/L)^{km}$. Given that we typically observe $m = 1.93$ tLOH events per clone with a median length of $s = 156.36$ kb, we estimate $P_{\text{overlap}} \approx 13\%$. We correct for such overlaps in the estimation of total iLOH rates by subtracting the length of the genome affected by tLOH events in each clone from the total genome length.

Any 2 tLOH events that occur on the same chromosome arm in the same MA line will necessarily overlap and will be detected as one. To correct for this undercount, we use the $P_0$ method to

estimate the tLOH event rates, as described in the Section "Calculation of LOH event and conversion rates."

## Calculation of LOH event and conversion rates

Our source data are the counts $C_{hca}$ of iLOH or tLOH events on each chromosome arm $a = 1, 2, ..., A$ in clone $c = 1, 2, ..., n_h$ of each hybrid $h = 1, 2, ..., H$. In our case, $A = 32$, $H = 12$, and the number of clones $n_h \in [10, 175]$ differs among hybrids.

*iLOH events.* We estimated the rate of iLOH events on chromosome arm $a$ in hybrid $h$ as $\hat{\lambda}_{ha} = \frac{\bar{C}_{ha}}{T_h}$, where $\bar{C}_{ha}$ is the average count of iLOH events on chromosome arm $a$ across all end-point clones of hybrid $h$ and where $T_h$ is the duration of the MA experiment in that hybrid. We estimate the total iLOH rate $\hat{\lambda}_h$ across the whole genome in hybrid $h$ as the sum of $\hat{\lambda}_{ha}$ across all chromosome arms. We then adjust this estimate using the correction for inter-marker LOH events described above. We do not correct for potential iLOH event overlaps, as these events are relatively short, making the effect likely negligible (see above). To correct the genome-wide iLOH event rate for possible overlaps with tLOH events, we divide the $\hat{\lambda}_h$ (after the inter-marker correction) by the fraction of the genome not affected by terminal events, averaged across all end-point clones.

*tLOH events.* Since we cannot observe more than one tLOH event in any given MA line, we estimate the tLOH rates using the $P_0$-method (Rosche and Foster 2000). We model the number of tLOH events occurring on chromosome arm $a$ in hybrid $h$ as a Poisson random variable with mean $\lambda_{ha}T_h$. Then, the probability that no events occur on this arm in a given MA line is $\exp(-\lambda_{ha}T_h)$. Thus, we estimate $\hat{\lambda}_{ha} = -\frac{\ln p_{0,ha}}{T_h}$ where $p_{0,ha} = \frac{1}{n_h+1}\left[\sum_{c=1}^{n_h}(1 - C_{hca}) + 1\right]$ is the fraction of end-point clones with no detected tLOH events on chromosome arm $a$, where we added a pseudocount 1. We estimate the total genome-wide rate of tLOH events by summing all $\hat{\lambda}_{ha}$ over all chromosome arms. Since, as mentioned above, all detected tLOH events are at least 7.5 kb long, and the expected between-marker distance in all of the hybrid studies here is less than 2 kb, the probability of tLOH events falling between markers is negligible (as long as they affect nontelomeric regions). Thus, we do apply the correction for between-marker events for tLOH events.

*Local LOH rates.* To estimate local LOH event rates, for each LOH event, we first define the position of the DNA break that caused it as the midpoint between the event boundaries (for all iLOH events and tLOH events shorter than 20 kb) or the position within the LOH 10 kb removed from the centromere-proximal event boundary (for tLOH events longer than 20 kb). We then divide each chromosome into nonoverlapping 50 kb windows (see Section "High-conversion high-bias (HCHB) windows") and count the number of breakpoints in each window across all end-point clones. We obtain the LOH event rate in each window by dividing these counts by the product of the window size, number of clones, and number of generations.

*Conversion rate.* We calculate the genome-wide LOH conversion rate for a given hybrid by summing the lengths of all events across all end-point clones and dividing this sum by the number of clones and effective length of the genome (see Section "Single-reference genotyping" for the definition of the effective length). We calculate tLOH conversion rates in the same way. For the iLOH conversion rate, we divide the summed lengths by the number of clones and the effective genome length minus the mean length of the genome masked by terminal events.

## Estimation of LOH detection errors
### High-conversion high-bias (HCHB) windows

To assess how the LOH conversion bias is distributed along the genome, we divide the genome into nonoverlapping windows of $\ell = 50$ kb length. If $L$ is the chromosome length, then we can fit $K = \lfloor L/\ell \rfloor$ full-length windows into the chromosome, where $\lfloor x \rfloor$ represents the integer part of $x$. We distribute these windows along the chromosome so that the $r = (L-K\ell)/2$ kilobases remain at each chromosome end. If $r > 25$ kb, then we form additional shorter windows at each chromosome end. If $r \leq 25$ kb, then we form larger windows at chromosome ends by merging these bases with the adjacent windows. For each window, we count the number of event breakpoints that convert toward each homolog (see Section "Calculation of LOH event and conversion rates" for how we define breakpoints). Assuming that LOH conversions occur with equal probability toward either of the 2 homologs and ignoring correlations between neighboring windows, we model these counts as binomial random variables with success probability $P = 0.5$. Thus, to identify windows with unexpectedly high conversion bias, we calculate the binomial $P$-value for the observed counts in each window and apply the Benjamini–Hochberg multiple testing correction (Benjamini and Hochberg 1995) with FDR $\leq 0.05$.

### Rough estimate of the number of erroneous LOH events detected with single-reference genotyping

We can roughly estimate the number of erroneous LOH events detected with the single-reference genotyping method if we assume that (i) there is no conversion bias and (ii) all erroneous LOH events convert toward the reference homolog. In our dataset, we detect 2,478 BY and 1,285 RM conversions against the S288C reference (which is very similar to BY). Under our assumptions, the number of RM-converted LOH events represents the true per-homolog LOH count, which implies that 1,193 out of 3,763 (31.7%) excess BY events represent false LOH detections. An analogous calculation with respect to the RM reference suggests that 462 out of 3,582 (12.9%) are erroneous.

### Estimation of LOH detection error rates from genotyping error rates

Here, we use our estimates of genotyping error rates to calculate the bounds on the numbers of missed and spurious LOH events identified by the single-reference genotyping procedure with the S288C genome used as reference. These estimates are not meant to be precise. Rather, our purpose for reporting these conservative estimates is to illustrate that even seemingly small error rates have the potential to substantially affect the inference of LOH events.

Each homozygous site that is falsely identified as heterozygous in an end-point clone (bottom left square in Fig. 2c) would lead us to miss either zero or one LOH event. Given that our estimated Hetero-FDR is $1.14 \times 10^{-4}$ per heterozygous site and the fact that a typical end-point clone has 29,687 apparent heterozygous markers, we would expect $29,687 \times 1.14 \times 10^{-4} = 3.38$ of them to be in fact homozygous. Depending on the genomic distribution of these errors relative to LOH events, they could lead us to miss on average between 0 and 3.35 LOH events per end-point clone, or up to $175 \times 3.35 = 592$ (15.8%) events total.

If a true homozygous site is falsely identified as heterozygous in a founder clone (top left square in Fig. 2c), it could give rise to multiple spurious LOH events, potentially in all of its descendant end-point clones. Given our estimated Hetero-FDR and the fact that a typical founder has 30,984 apparently heterozygous markers, we expect $30,984 \times 1.14 \times 10^{-4} = 3.53$ of them to be in fact homozygous. These errors could lead us to identify between 0 and 3.53 spurious LOH events per end-point clone or up to $175 \times 3.53 = 618$ (16.4%) total. Moreover, whenever a false heterozygous error occurs in a founder, all the resulting spurious LOH events would appear to be converted to the same homolog, potentially explaining the existence of the HCHB sites discussed above.

Each heterozygous site that is falsely identified as homozygous in an end-point clone (bottom right square in Fig. 2c) would give rise to zero or one spurious LOH event in that clone. Given that we estimated Homo-FDR to be 0.062 per homozygous site and the fact that a typical end-point clone has 1,597 apparently homozygous sites, we expect 99.0 of them to be in fact heterozygous. In the best-case scenario, all of these errors are concentrated in the same genomic region, which underwent an actual LOH event that is supported by additional markers. In this case, they do not inflate our estimate of the LOH rate. In the worst-case scenario, the 99.0 expected false homozygous sites are distributed broadly across the genome and contribute to many spurious LOH events. If a typical end-point clone had more than 99 LOH events supported by a single marker, then the maximum number of false LOH events would be 99. In reality, a typical end-point clone only has a total of 21.2 LOH events. If the 99.0 expected false homozygous sites instead support those events with the fewest markers, they would produce 19.2 (90%) false LOH events in a typical clone, or 3,353 events total.

Finally, if a true heterozygous site is falsely identified as homozygous in a founder clone (top right square in Fig. 2c), it reduces our power to detect LOH events that may occur in the vicinity of that marker in the descendants of that founder. Given the estimated Homo-FDR, we expect that 25 out of 403 sites called homozygous in a typical founder are in fact heterozygous. Since a typical founder has over 30 thousand heterozygous sites, this reduction in power is negligible.

Combining all of these estimates together, we conclude that between 0 and 592 (15.8%) of LOH events may have been missed and between 0 and $618 + 3,353 = 3,971$ (105%) may be spurious.

## Analysis of variance in the LOH event rates

To assess the relative impact of cis- and trans-acting genetic variation on the LOH rate, we constructed a series of nested generalized linear mixed effects models (GLMMs). Our original data are the counts $C_{hca}$ of iLOH or tLOH events on each chromosome arm $a = 1, 2, \ldots, A$ in clone $c = 1, 2, \ldots, n_h$ of each hybrid $h = 1, 2, \ldots, H$ (Supplementary Table 4 in Supplementary File 2). In our case, $A = 32$, $H = 12$, and the number of clones $n_h \in [10, 175]$ differs among hybrids. We model the per-generation rate of LOH events using equation (6). In this equation, $\mu$ and $\alpha_h$ are fixed-effects coefficients and $\beta_a$ and $\gamma_{ha}$ are random-effects coefficients, which are assumed to be distributed normally with zero mean and standard deviations $\sigma_{fc}$ and $\sigma_{sc}$, respectively. The term $\mu + \alpha_h L_a$ represents hybrid-specific differences in the rate of LOH events that are uniformly distributed along the entire genome. We refer to this term as "segregating trans factors" and use index "st" to refer to the variance attributed to it. The term $\beta_a$ represents chromosome-arm-specific effects that are shared among all hybrids. We refer to this term as "fixed cis factors" ("fixed" in the population genetics sense, i.e., not segregating) and use the index

"fc" to refer to variance attributed to it. Finally, the term $\gamma_{ha}$ represents hybrid- and chromosome-arm-specific differences in LOH rates. We refer to this term as "segregating cis factors" and use the index "sc" to refer to variance attributed to it. We refer to model (6) that contains all 4 terms as the "Full model". Below, we also consider a series of models nested within it that contain only subsets of these terms.

Since all MA lines from hybrid $h$ have been propagated for $T_h$ generations, the expected number of LOH events that have occurred on chromosome arm $a$ on the line of descent of clone $c$ of hybrid $h$ is given by $\xi_{hca} = \lambda_{hca} T_h$. Since our ability to observe multiple LOH and LOH events per chromosome arm in the same clone differs between interstitial and terminal events (we can only observe at most one tLOH event per arm per clone but many iLOH events), we model iLOH and tLOH events differently.

### iLOH events

We model the number $C_{hca}$ of observed iLOH events as a Poisson random variable with the expectation $\mu_{hca} = f(\xi_{hca})$, where $\xi_{hca} = \lambda_{hca} T_h$ and $f = g^{-1}$ is the inverse of the link function $g$, which we define as $g(y) = y$ when $y \geq \delta$, and $g(y) = \delta + \delta \ln y/\delta$ when $0 < y < \delta$. In other words, $g$ is an identity function that is smoothly stitched to a logarithmic function at the threshold value $\delta$. This link function ensures that the expected number of iLOH events consists of the components shown in equation (6), as long as the predictor $\xi_{hca}$ exceeds the threshold $\delta$; at the same time, this link function prevents the Poisson expectation from turning negative when the predictor $\xi_{hca}$ becomes negative. We set $\delta = 0.01$, which is the smallest value at which the fitting procedure for all of our models reliably converged.

*Model selection.* We carried out model selection by constructing a series of models nested within the Full model (6) and using the likelihood ratio test. To assess the significance of the segregating cis effects, we compared the Full model with the model

$$\lambda_{hca} = \mu + \alpha_h L_a + \beta_a \tag{4}$$

which we refer to as the "Arm" model. To assess the significance of the fixed cis effects, we compare the Arm model with the model

$$\lambda_{hca} = \mu + \alpha_h L_a \tag{5}$$

which we refer to as the "Hybrid-slopes" model. The results of these comparisons are provided in Supplementary Table 5 in Supplementary File 2. Since the Full model is statistically better than the Arm model, we use it to describe the distribution of iLOH events. The best-fit parameters for the Full model are reported in Supplementary Table 6 in Supplementary File 2. We note that all fixed-effect predictors $\hat{\mu} + \hat{\alpha}_h L_a$ exceeded $\delta = 0.01$, and only 1.02% of all $\hat{\lambda}_{hca}$ predicted values are below $\delta$, suggesting that our chosen threshold value did not have a strong effect on the inference.

*Variance partitioning.* We estimate the amount of residual variance $V_{noise}$ (i.e. not explained by the Full model) as the mean squared difference between $C_{hca}$ and the predicted values $f(\hat{\lambda}_{hca} T_h)$, where $f = g^{-1}$ is the inverse of the link function described above. We estimate the amount of variance $V_{cis + noise}$ that is attributed to all cis factors and measurement noise (i.e. not explained by fixed effects) as the mean squared difference between $C_{hca}$ and the predicted values $f(\hat{\mu} T_h + \hat{\alpha}_h L_a T_h)$. Then, the amount of variance attributed to the segregating trans factors is

$V_{st} = V_{tot} - V_{cis + noise}$, where $V_{tot}$ is the total variance of $C_{hca}$. We estimate the amount of variance $V_{sc + noise}$ that is attributed to segregating cis factors and noise (i.e. unexplained by the first 3 terms in equation (6)) as the mean squared difference between $C_{hca}$ and $f(\hat{\mu}T_h + \hat{\alpha}_h L_a T_h + \hat{\beta}_a T_h)$. Then, the amount of variance attributed to the fixed cis factors is $V_{fc} = V_{cis + noise} - V_{sc + noise}$ and the amount of variance attributed to the segregating cis factors is $V_{sc} = V_{sc + noise} - V_{noise}$. The fractions of explained variance are obtained from these values by dividing them by $V_{tot}$.

### tLOH events

Since we can only observe at most one tLOH event on a given chromosome arm of a given end-point clone, we model the number $C_{hca}$ of the observed tLOH events as a Bernoulli random variable with the success probability $p_{hca} = f(\xi_{hca})$, where $f = g^{-1}$ is the inverse of the link function $g$, which we define as $g(y) = -\log(1-y)$ when $\epsilon \leq y < 1$ and $g(y) = \delta + \epsilon e^\delta \ln\frac{y}{\epsilon}$ when $0 < y < \epsilon$, with $\epsilon = 1 - e^{-\delta}$. With this link function, $p_{hca} = 1 - \exp(-\xi_{hca})$ as long as $\xi_{hca} \geq \delta$, i.e. $p_{hca}$ equals the probability that at least one tLOH event will be observed from the Poisson process with rate $\lambda_{hca}$. When $\xi_{hca} < \delta$, we have $p_{hca} = \epsilon \exp\frac{\xi_{hca} - \delta}{\epsilon e^\delta}$, which ensures that $p_{hca}$ remains positive for any predictor value. For this series of models, we set $\delta = 0.03$.

*Model selection.* We carried out model selection for tLOH events analogously to iLOH events, with the following differences. After fitting the Full model, we found that the estimates of $\sigma_{fc}$ and $\sigma_{sc}$ are zero within our numerical tolerance, implying that the fitted Full model is equivalent to the Hybrid-slopes model (equation (5)). We therefore decided to assess the significance of segregating cis effects by comparing the model $\lambda_{hca} = \mu + \gamma_{ha}$, which we refer to as the "Hybrid-Arm" model, against the null model without these effects. We also found no support for the Hybrid-Arm model. Then, we assessed whether the trans factors affecting the rate of tLOH events are segregating (i.e. variable across hybrids) or fixed (i.e. shared across hybrids). To this end, we compared the Hybrid-slopes model against the "One-slope" model $\lambda_{hca} = \mu + \alpha L_a$. The results of these model comparisons are provided in Supplementary Table 5 in Supplementary File 2. We find that the Hybrid-slopes model is statistically better than the One-slope model. We therefore report the best-fit parameters for the Hybrid-slopes model in Supplementary Table 6 in Supplementary File 2. We note that the fraction of fixed-effect predictors $\hat{\mu} + \hat{\alpha}_h L_a$ that fell below $\delta = 0.03$ was 12.0%. However, when we fitted the Hybrid-slopes model with the reduced threshold of $\delta = 0.025$, the parameter estimates remained essentially unchanged, whereas the fraction of fixed-effects predictors below the threshold dropped to 2.86%. This again suggests that our chosen threshold value did not have a strong effect on the inference.

*Variance partitioning.* We partition the variance for tLOH events analogously to iLOH events. Specifically, we estimate the amount of residual variance $V_{noise}$ (i.e. not explained by the Hybrid-slopes model) as the mean squared difference between $C_{hca}$ and the predicted values $f(\hat{\mu}T_h + \hat{\alpha}_h L_a T_h)$. The amount of variance attributed to the segregating trans factors is $V_{st} = V_{tot} - V_{noise}$.

*Analysis of tLOH events on the longest chromosome arms.* We selected chromosome arms with a length exceeding 0.5 Mb. These are Chrs. IV-r (1.08 Mb), XII-r (0.93 Mb), XV-r (0.76 Mb), XIII-r (0.66 Mb), XIV-l (0.63 Mb), VII-r (0.59 Mb), II-r (0.57 Mb), and XVI-l (0.56 Mb). For each of these arms, we carried out the G-test of homogeneity using the $\chi^2$ distribution with 11 degrees of freedom to test whether the proportion of tLOH events that occur on that arm is the same in all hybrids. The P-values were adjusted using the Bonferroni multiple testing correction.

### Implementation

We fit all generalized linear mixed effects models using the fitglme function in MATLAB R2024b with the option of approximating the log-likelihood using Laplace's method. We fit all generalized linear models using the function fitglm. We obtain the best-fit predicted values $f(\hat{\lambda}_{hca}T_h)$ using the predict function applied to the appropriate model.

## Results
### Standard LOH detection is prone to genotyping errors

As mentioned in the Introduction, previous studies that characterized LOH rates and distributions used various, often custom-built, pipelines for detecting LOH events, which all rely on single-reference genotyping (Lee et al. 2009; St Charles et al. 2012; Yim et al. 2014; Laureau et al. 2016; Dutta et al. 2017, 2021; James et al. 2019; Tattini et al. 2019; Loeillet et al. 2020; Pankajam et al. 2020; Sui et al. 2020; Mozzachiodi et al. 2021; Vijayan et al. 2025). We have implemented our own version of this approach (see Section "Single-reference genotyping" in Materials and Methods). For simplicity, we refer to this single-reference genotyping method as "standard." We begin our investigation by demonstrating that this standard method is susceptible to genotyping errors, which lead to inaccurate LOH inferences. To do so, we analyze the genomic data from our MA experiment in yeast described in Overton et al. (2023). Briefly, in this experiment, we crossed two haploid yeast strains, BY and RM, to produce a number of diploid "founder" clones that are heterozygous at approximately 40,000 sites across the genome. Our experiment was originally designed to test the effects of a gene-drive element on LOH rates; therefore, each founder carries 1 of 3 homozygous genetic constructs at the ADE2 locus. Each MA line was propagated for about 800 generations, and we sequenced both the founder and the endpoint clones to a median depth of 36.7×. Since we found only minimal differences in the number of accumulated LOH events between gene-drive strains (Overton et al. 2023), here we pool the data from 175 MA lines in our experiment (see Section "Strains and media" in Materials and Methods for details), which yields the largest dataset for LOH detection for any single hybrid.

To detect LOH events that may have occurred during our MA experiment, we used the standard method with the *Saccharomyces cerevisiae* S288C as the reference genome sequence (see Section "Single-reference genotyping" in Materials and Methods). We identified 3,763 LOH events across 175 end-point clones (rate $2.7 \times 10^{-2}$ generation$^{-1}$) supported by 40,485 converted markers. Surprisingly, we found that 2,479 (65.9%) of these events apparently converted initially heterozygous markers to the BY homolog (Fig. 1a), which is far in excess of 50% expected under the null hypothesis of no homolog bias (P-value = $2.4 \times 10^{-86}$, binomial test). We found similar homolog bias in previous studies where it was measured, though not always explicitly reported (Loeillet et al. 2020; Pankajam et al. 2020). While some authors attributed this bias to biological sources (Loeillet et al. 2020), we hypothesized that it may be at least in part driven by genotyping errors.

To probe this hypothesis, we investigated the distribution of detected LOH events and their homolog bias across the genome. We divided the genome into 50-kb windows and calculated the

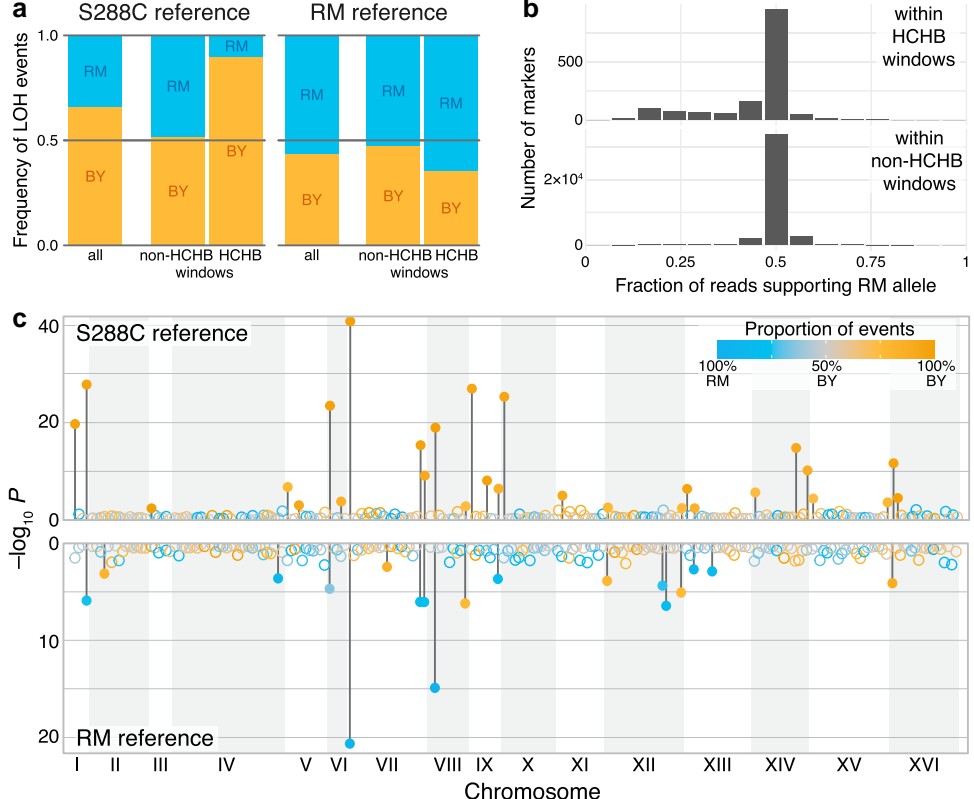

**Fig. 1.** Single-reference genotyping can result in LOH events biased toward the reference homolog. a) Fraction of LOH events converting toward each homolog, when either the S288C (a close relative of our BY strain) or the RM genomes are used as reference, as indicated (see Section "Dual-reference genotyping" in Materials and Methods for details). Each bar shows the frequency of LOH events converting toward the BY (orange) or RM (blue) homolog. Left, middle, and right bars in each subplot represent all LOH events, those with breakpoints located within non-HCHB or HCHB windows, respectively. b) Distribution of RM allele support among heterozygous markers within HCHB and non-HCHB windows, when the S288C genome is used as reference. c) Distribution of HCHB windows across the yeast genome when S288C (top) or RM (bottom) genomes are used as reference. Each point represents a 50-kb window in the yeast genome. The y-axis shows the binomial P-value for homolog bias of the LOH events with breakpoints within each window. The color of each point indicates the fraction of LOH events converting to each homolog, as indicated in the legend. Filled circles indicate HCHB windows, i.e., windows with a significant bias after the Benjamini–Hochberg multiple testing correction with FDR ≤ 0.05; non-HCHB windows are shown as empty circles.

binomial P-value for the observed homolog bias for each window (see Section "High-conversion high-bias (HCHB) windows" in Materials and Methods). We found that 28 out of 228 genomic windows had a conversion bias that was statistically significant at $P < 0.05$ after the Benjamini–Hochberg multiple testing correction (Fig. 1a), and we refer to these windows as highly converted, highly biased (HCHB). All 28 HCHB windows exhibited a bias toward BY because 1,503 out of the 1,670 (90%) LOH events that overlapped with these windows favored the BY homolog. In contrast, only 1,001 out of 1,936 (52%) of LOH events that overlapped with non-HCHB windows favored the BY homolog. This suggests that the observed homolog bias in LOH events is driven by local genomic regions highly enriched with apparent BY conversions.

One possible reason for these localized biases is that the genotyping quality is worse within HCHB windows than elsewhere. If this is the case, signatures of bias may be detectable even among markers that are called heterozygous. Thus, we investigated the distribution of reads supporting each parental allele among heterozygous markers within HCHB and non-HCHB windows (Fig. 1b). We found that the read support distribution at markers within HCHB windows was wide (SD = 0.12) and highly skewed toward the reference allele (skewness = −0.56) compared to a rather narrow (SD = 0.10) and fairly symmetric (skewness = −0.028) distribution at heterozygous markers within non-HCHB windows,

resulting in a significant difference between the two ($P < 0.01$, permutation test for skewness).

Finally, if the LOH conversion biases arise largely due to genotyping errors, we would expect that the sign of the conversion bias would flip and the genomic distribution of HCHB windows would change if we use the other parental genome as a reference. After genotyping our clones with respect to the RM reference, we identified 3,582 LOH events supported by 40,652 converted markers, 2,022 (56.4%) of which now favored the RM homolog ($P = 6.1 \times 10^{-15}$, binomial test; Fig. 1a). This conversion bias was driven by 18 HCHB windows (Fig. 1a and c), 12 of which now favored the RM homolog. Out of all HCHB windows we found in the S288C and RM datasets, 16 were found only in the S288C dataset, and 6 only in the RM dataset. Even for those 12 windows that contained significant conversion biases in both call sets, 8 had biases that favored the homolog of the parental reference (Supplementary Fig. 1 in Supplementary File 1). We note that the parental BY strain in our experiment is closely related to but is not identical to S288C. However, repeating the single-reference genotyping and analyses using the BY parental reference genome did not qualitatively change the level of LOH biases we observe.

Taken together, these results indicate that the observed LOH event biases are at least in part driven by genotyping errors, many of which are concentrated within HCHB windows. We can

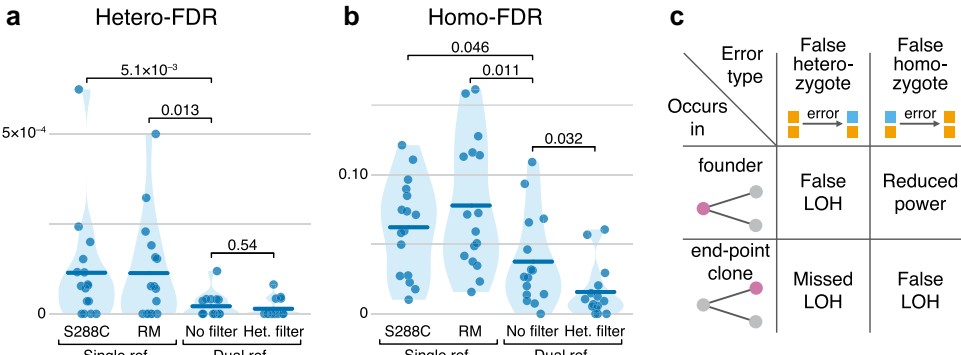

**Fig. 2.** Genotyping errors and their effects on inferred LOH events. a) Estimated rate of false heterozygous calls in our data (Overton et al. 2023) across different genotyping procedures, as indicated. See Section "Genotyping" in Materials and Methods for details. Each point represents a family, i.e., all end-point clones descendant from the same ancestor. Thick horizontal lines represent the means. P-values on top are from permutation tests of the means. b) Estimated rate of false homozygous calls. c) Potential effects of different genotyping errors on the inference of LOH events.

crudely estimate how such errors could potentially affect our LOH estimates by assuming that all LOH events that convert toward the nonreference homolog are correct and all "excess" conversions toward the reference homolog are erroneous (see Section "Rough estimate of the number of erroneous LOH events detected with single-reference genotyping" in Materials and Methods for details). According to this estimate, in our dataset, a failure to control for genotyping errors and/or biases could result in an LOH rate estimate that exceeds the true rate by as much as 31.7%. We note, however, that the magnitude of this error could be much higher or much lower in other datasets, depending on various parameters, such as the actual number of observed LOH events, the number of sequenced clones, etc.

## Quantification of genotyping errors

We next sought to leverage our experimental design to quantify genotyping errors more directly. In our experiment, 175 MA lines form 16 "families" as each line is descendent from one of 16 "founder" strains. As mentioned above, we sequenced all founder and end-point clones to the same average coverage (see Section "Illumina sequencing" in Materials and Methods). Thus, we can infer the genotype of every founder either from its own sequencing data or from the majority of the genotypes of its descendants. Comparing these two calls allows us to estimate the false heterozygous discovery rate ("Hetero-FDR"), i.e. the fraction of sites that are in fact homozygous among those identified as heterozygous, and the false homozygous discovery rate ("Homo-FDR"), i.e. the fraction of sites that are in fact heterozygous among those identified as homozygous.

To this end, we developed a model for calculating the odds ratio for a marker site in a founder to be false homozygous or false heterozygous depending on the configuration of genotypes of its end-point clones (see Section "Estimation of genotype call error rates" in Materials and Methods). Our model depends on 5 parameters, 3 of which we are able to constrain using prior knowledge. The 2 remaining unconstrained parameters are the false homozygote probability $\epsilon_{HOM}$, i.e. the probability that a clone is called homozygous at a site where it is in fact heterozygous, and the false heterozygote probability $\epsilon_{HET}$, i.e. the probability that a clone is called heterozygous at a site where it is in fact homozygous. We varied these unknown probabilities over 4 orders of magnitude between $10^{-4}$ and 0.1 and found that in this entire parameter space, the founder's genotype call is likely erroneous if the genotype calls of all of its descendant end-point clones unanimously disagree

with it, provided that at least 6 descendant clones have been genotyped (Supplementary Fig. 2 in Supplementary File 1). Considering such sites as erroneous, we estimate the Hetero-FDR to be $1.14 \times 10^{-4}$ and $1.13 \times 10^{-4}$ per heterozygous call for the single-reference genotyping procedure which uses S288C and RM genomes are reference (Fig. 2a; Supplementary Table 1 in Supplementary File 2), respectively; and we estimated the Homo-FDRs to be 0.062 and 0.078 per homozygous call, respectively (Fig. 2b). Note that this high Homo-FDR is consistent with a previous estimate (Bobo et al. 2016). Importantly, it does not imply that a large fraction of all sequenced homozygous sites are genotyped incorrectly (indeed, most of the genome is shared by both parents of all clones). Rather, the reference set for Homo-FDR is the set of sites identified as homozygous among those that are polymorphic within each family (see Section "Genotyping" in Materials and Methods).

These two types of genotyping errors have different implications for the detection of LOH events, depending on whether they occur in the founder or end-point clones (Fig. 2c). We can use our error estimates to obtain bounds on the numbers of missed and spurious LOH events (see Section "Estimation of LOH detection errors" in Materials and Methods for details). We find that between 0 and 586 LOH events, which constitute up to 15.8% of all LOH events detected with the single-reference method, may have been missed and between 0 and 3,763 (100%) of the detected events may be spurious, depending on the genomic distribution of genotyping errors relative to true LOH events. The upper bounds reported here are conservative, and the numbers of erroneous LOH detections lie likely somewhere between the bounds. Nonetheless, these estimates offer an additional confirmation that genotyping errors have the potential to strongly affect the inference of LOH events. More importantly, the procedure for estimating genotyping error developed here allows us to quantitatively evaluate the performance of any improved genotyping method on our dataset.

## An improved genotyping method

Next, we sought to develop a genotyping approach with reduced error rates that would enable us to more accurately identify LOH events and estimate their rates. Our approach improves upon the existing ones in 3 key ways. First, it is symmetric with respect to the two parental reference genomes, which should reduce genotyping biases. Second, we introduce a new statistically grounded filter for removing dubious heterozygous calls.

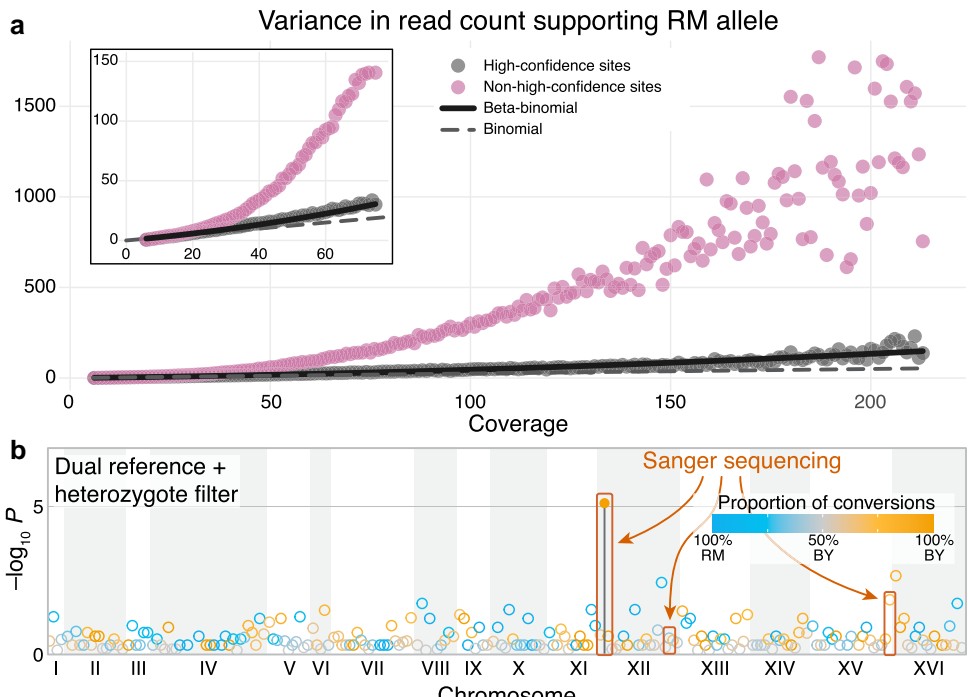

**Fig. 3.** Heterozygous genotype filter. a) The variance in the read count supporting the RM allele for high-confidence heterozygous sites (gray points) and non-high-confidence heterozygous sites (red points). The beta-binomial model (black line) with $\rho = 2.09 \times 10^{-3}$ fits the high-confidence data well (see text for details). The inset shows an enlarged version of the data at low coverages. b) Only one HCHB window remains when we use our improved genotyping procedure to detect LOH events. Regions that we validated with Sanger sequencing are shown.

While filtering out dubious homozygous calls would also be desirable, implementing such a filter is difficult (see "Discussion"). Finally, we develop corrections to account for unobserved LOH events.

### Dual-reference genotyping

As we have shown above, the choice of the reference genome can result in genotyping errors, likely due to biases during read mapping. Thus, we reasoned that combining genotype-call information derived from both parental references might reduce both the incidence of these errors and genotyping biases. We devised a procedure—described in detail in Section "Dual-reference genotyping" in Materials and Methods—that implements this idea. Briefly, for each clone, we first apply a standard single-reference genotyping pipeline twice, once with respect to each parental reference genome. We thereby obtain two genotype calls per site. If both calls are of low quality or if 1 of the 2 genotypes is missing, the site is discarded because it cannot be genotyped reliably. If the two calls agree (and at least one of them is of high quality), there is no ambiguity. If the calls disagree and one of them has much higher quality than the other, we retain this call and discard the lower-quality one. If the calls disagree and both are of high quality, we discard the site.

Applying the single-reference pipeline, we made 7,865,725 and 8,002,783 genotype calls with respect to the S288C and RM references across all clones, respectively. 492,360 (6.26%) and 629,418 (7.87%) of these calls were specific to the S288C and RM reference, respectively, and were discarded, as were 618,768 calls that were of low quality with respect to both references. The remaining 6,754,597 calls supported by both references were distributed across 41,884 sites. 6,581,004 (97.43%) of them were concordant between the two references. The remaining 173,593 (2.57%) discordant calls were broadly distributed across 22,352 (53%) sites and were reconciled as described above. In particular, 137,486

(79.2%) of the discordant calls were homozygous for the reference allele in one callset and heterozygous in the other, with the heterozygous call being of substantially higher quality in 118,362 (86%) of them, which is consistent with our relatively high estimate of the Homo-FDR among polymorphic sites (see Fig. 2b). From this reconciled callset, we obtained the set of 5,974,454 high-quality calls across 37,716 marker sites.

Despite the fact that discordant sites constitute a small fraction of all genotyped sites, our dual-reference reconciliation procedure significantly reduces both genotyping errors (Fig. 2a and b; Supplementary Table 1 in Supplementary File 2) and the prevalence of HCHB windows (Supplementary Fig. 3 in Supplementary File 1). Specifically, the Hetero-FDR fell more than 6-fold to $2.09 \times 10^{-5}$ per heterozygous site ($P = 5.1 \times 10^{-3}$, permutation test across families), and the Homo-FDR fell almost by half to $3.76 \times 10^{-2}$ per homozygous site ($P = 0.046$).

### Filtering of spurious heterozygous calls

Falsely identified heterozygous sites could be an important source of errors that lead to the inference of false reference-biased LOH events (see Section "Estimation of LOH detection errors" in Materials and Methods for details). Thus, we sought to develop a procedure for filtering out dubious heterozygous calls. To do so, we first identified a set of high-confidence heterozygous sites in our founder clones as those that were called heterozygous in our dual-reference procedure and were also supported by heterozygous calls in at least 2/3 of the descendant clones (see Section "Heterozygous genotype filter" in Materials and Methods for details). We then investigated the distributions of reads supporting the RM allele at these high-confidence sites. As expected, the average number of such reads was very close to $n/2$ where $n$ is the total coverage for the site (Supplementary Fig. 4a in Supplementary File 1). In the simplest model, the number of reads supporting one

allele would follow the binomial distribution with success probability 50%, such that the variance around this mean would be $n/4$. In fact, we find that the variance increases faster than this theoretical expectation even at high-confidence heterozygous sites (Fig. 3a, gray points). Higher variance could arise if the sequenced alleles are drawn from a pool in which the allele frequencies randomly deviate from 50% (e.g. due to noise arising during sequencing library preparation). To account for this excess variance, we model the number of reads supporting the RM allele as a beta-binomial random variable whose variance is $n/4 + \rho(n^2 - n)$, where parameter $\rho$ controls the excess variance. We find that this model with $\rho = 2.09 \times 10^{-3}$ provides an excellent fit to our data (Fig. 3a and Supplementary Fig. 5 in Supplementary File 1; $R^2 = 0.90$, $P < 0.01$).

We next interrogated the distributions of reads supporting the RM allele at non-high-confidence sites. In contrast to high-confidence sites, the average number of such reads at non-high-confidence sites deviates slightly from $n/2$ at higher read depths (Supplementary Fig. 4b in Supplementary File 1), suggesting that a small residual bias favoring the BY reference remains despite our best efforts to eliminate it. More importantly, the variance in RM allele count at these sites far exceeds the variance at high-confidence sites (Fig. 3a and Supplementary Fig. 4c in Supplementary File 1), which suggests that many of the non-high-confidence heterozygous calls may be dubious, especially at high-coverage sites. To eliminate such potentially dubious calls, we calculated the 95% C.I.s for the RM allele count at each read depth $n$ based on the beta-binomial model described above and removed all heterozygous calls whose alternative read support fell outside of these intervals (see Section "Heterozygous genotype filter" in Materials and Methods for details).

This filtering procedure removed 220,618 out of 6,386,206 (3.5%) heterozygous calls with the most extreme RM allele frequencies. As a result, the Hetero-FDR dropped further, though not significantly so, to $1.37 \times 10^{-5}$ per heterozygous call (Fig. 2a; Supplementary Table 1 in Supplementary File 2) and only one HCHB window remained (Fig. 3b). The removal of dubious heterozygous calls also further reduced our estimate of Homo-FDR to $1.57 \times 10^{-2}$ per homozygous call ($P = 0.032$; Fig. 2b), which could be caused by the reduction in the number of sites where at least 6 end-point clones were genotyped and called heterozygous.

## Recalcitrant HCHB markers can arise from structural variation

Our improved genotyping procedure eliminated all HCHB windows except one on the left arm of Chr XII (Fig. 3b). To investigate whether this region harbors another unaccounted type of genotyping error or possibly a real LOH hotspot, we designed PCR primers that uniquely targeted a 489 bp fragment that contains 5 HCHB markers within this region. We amplified this fragment from the genomic DNA of the haploid BY and RM parent strains, 2 diploid founder clones and 7 descendant end-point clones, 6 with apparent conversions and one without. We found that the BY and RM parents carry 3 and 5 81 bp tandem repeats at this locus, respectively, and RM also has 5 single-nucleotide variants (SNVs) in one of the repeats (Supplementary Fig. 6 in Supplementary File 1). Consequently, the PCR amplification of this locus in all of our BY/RM diploids yielded 2 fragments of lengths 489 bp and approximately 700 bp corresponding to the BY and RM alleles. We Sanger sequenced these 2 fragments and found that the sequence of the shorter (BY) fragment corresponds to the BY reference in all sequenced clones, as expected. However, unexpectedly, the longer (RM) fragment was heterozygous for the

5 SNVs in all clones (Supplementary Fig. 6 in Supplementary File 1, Supplementary Table 2 in Supplementary File 2, and Supplementary Data 1), implicating either PCR artifacts or recombination events during the construction of founder strains. Regardless of the causes, there was no support for LOH events identified at these markers in the short-read data. Instead, Sanger sequencing suggests that accurate short-read-based genotype inference at this locus is challenging due to the structural variation in the parents. Therefore, we removed these 5 markers from further analyses.

## Validation of the genotyping procedure

To validate our genotyping procedure and the quality of the inferred LOH events, we Sanger sequenced 2 founders and 7 end-point clones at 2 regions with high conversion rates: a 766-bp region on Chr XII at positions 956964–957730 containing 4 putative markers and a 869-bp region on Chr XV at positions 1030968–1031837 containing 6 markers (Fig. 3b). For both regions, we found that the Sanger sequencing genotypes matched those obtained by our revised pipeline across all 10 markers in all 9 sampled clones.

## Corrections for unobserved LOH events

Since our ability to detect LOH events depends on the density of initially heterozygous markers, we cannot observe all LOH events that occur in the MA experiment. There are 4 reasons for such undercounts.

First, we cannot detect LOH events that do not affect any of our marker sites. Shorter LOH events are more likely to fall between markers, which leads not only to an undercount of LOH events but also to a bias in the distribution of their lengths. We use the following simple correction to reduce (but not completely eliminate) this bias. Assuming that $n$ markers are uniformly distributed across a genome of length $L$, the probability that an LOH event of length $\ell$ converts at least one marker and is therefore detected is $P_{\text{detect}}(\ell) = 1 - (1 - \ell/L)^n$. Thus, if we detect $k_\ell$ LOH events of length $\ell$, the expected total number of LOH events of this length is $k_\ell/P_{\text{detect}}(\ell)$. Note that $P_{\text{detect}}(\ell)$ quickly approaches 1 for LOH lengths exceeding the average distance between the markers, $L/n = 321$ bp, as expected. We apply this correction only to iLOH events, since all of our tLOH events are longer than 7.5 kb and therefore never fall between markers (see Section "Corrections for undetected LOH events" in Materials and Methods).

Second, we can never resolve more than one tLOH event on the same chromosome arm. We correct for this undercount by estimating the rate of tLOH events using the analog of the $P_0$-method for the estimation of mutation rates (Rosche and Foster 2000) (see Section "Calculation of LOH event and conversion rates" in the Materials and Methods for details).

Third, two or more overlapping LOH events are observed as one event. We estimate that in our MA experiment, the probability for 2 iLOH events occurring in the same clone to overlap is 0.46%. Thus, we ignore such overlaps. On the other hand, the probability that at least one iLOH event overlaps with a tLOH event is about 13%. While we cannot resolve such events, we can account for such overlaps by adjusting the denominator in the calculation of iLOH rates (see Section "Corrections for undetected LOH events" in the Materials and Methods for details).

Finally, we cannot detect any LOH events that fall within repeat and telomeric regions of chromosomes because these regions are difficult to map reliably (see Section "Single-reference genotyping" in Materials and Methods). We account for it as above by further adjusting the denominator in the calculation of iLOH rates.

Thus, all of our results pertain to nontelomeric and nonrepeat regions of the genome.

## Improved genotyping significantly changes the number and characteristics of detected LOH events across yeast hybrids

Having demonstrated that our improved genotyping method substantially reduces the prevalence of genotyping errors and HCHB regions, we next sought to understand how these improvements affect the LOH rate estimates and the characteristics of the detected LOH events. We do so in two ways. First, we use our data to directly compare the rates and properties of LOH events identified by the standard and the improved pipelines. Second, we reanalyze several previously published datasets in various yeast hybrids and report the revised LOH rates and their properties.

### Direct comparison of LOH rates and characteristics in our dataset

*LOH counts and rates.* Using our improved genotyping pipeline, we detect a total of 1,288 LOH events across the 175 MA lines (Supplementary Data 2), an almost 3-fold reduction compared to 3,763 events detected by the standard pipeline ($P < 1 \times 10^{-4}$, permutation test, Supplementary Table 3 in Supplementary File 2). Out of 1,288 detected LOH events, 954 (74%) are interstitial and 334 (26%) are terminal, compared to 3,261 (87%) and 502 (13%) out of 3,763 detected with the standard method, respectively.

The absolute numbers of detected LOH events depend on several technical parameters of the study, such as the duration of the MA experiment and the number of heterozygous markers. For example, the standard pipeline applied to our data uses 40,485 markers, but only 37,716 of them (7.23% less) pass the more stringent filters of our improved pipeline, which could at least in part explain the reduction in the absolute number of detected LOH events. After applying our corrections for undetected events, we estimate that a total of 1,795 events with potentially detectable lengths have occurred in our experiment, with 1,435 (79.9%) of them being interstitial and 360 (20.1%) being terminal (Fig. 4b). These estimates are much smaller than those estimated based on the standard pipeline (total = 7,975; interstitial = 7,451 (93.4%); terminal = 524 (6.6%)). Thus, the reduction in the number of LOH events (especially interstitial) detected by our improved pipeline is not merely due to the reduction in markers. Rather, the standard method detects many more short poorly supported iLOH events, many of which are likely erroneous (Fig. 4c and Supplementary Fig. 7 in Supplementary File 1). Similarly, the improved method eliminates many of the short tLOH events detected by the standard method and supported by single markers near chromosome ends.

Next, we compute the LOH event and conversion rates. In contrast to absolute counts, these rates do not depend on technical parameters and can be compared across studies. As mentioned above, the LOH event rate is the per-generation rate at which LOH events occur, and the LOH conversion rate is the per-generation rate at which heterozygosity at a typical genomic position is lost due to LOH events. Based on the LOH counts corrected for undetected events, we estimate the overall LOH event rate to be $1.3 \times 10^{-2}$ per genome per generation (SEM = $4.5 \times 10^{-4}$) which is over 4 times lower than the rate of $5.6 \times 10^{-2}$ per genome per generation (SEM = $1.6 \times 10^{-3}$) estimated using the standard genotyping method ($P < 1 \times 10^{-4}$, permutation test; Fig. 4a). We estimate the LOH conversion rate to be $5.9 \times 10^{-5}$ per basepair per generation (SEM = $3.3 \times 10^{-6}$), which is about 4.4% lower than the $6.2 \times 10^{-5}$ per basepair per generation (SEM = $3.4 \times 10^{-6}$) rate

estimated with the standard method (Fig. 4a). As expected, the reduction in the LOH conversion rate is much smaller than the reduction in the event rate because our improved genotyping method filters out primarily short iLOH and tLOH events, which are responsible for a minor fraction of converted positions.

*Distribution across the genome.* The distribution of LOH events is more uniform across the genome after the revision. Figure 4d shows the LOH event rate computed for each 50 kb window in the yeast genome. We find that the SD of the distribution of local LOH event rates declines from $1.5 \times 10^{-9}$ bp$^{-1}$ generation$^{-1}$ for the standard method to $2.7 \times 10^{-10}$ bp$^{-1}$ generation$^{-1}$ for the revised method. Moreover, the number of windows with rates 3 times the median decreases from 24 to 14.

*Homolog bias.* We find that 661 (51.3%) out of 1,288 detected LOH events converted toward the BY homolog, which is not significantly different from the binomial expectation ($P = 0.20$), in contrast to a strong and highly significant 65.9% BY-bias among LOH events detected by the standard pipeline. We find the same trend when we separate iLOH and tLOH events, although the bias among iLOH events after applying the revised method remains marginally significant ($P = 0.024$; Supplementary Table 3 in Supplementary File 2).

*Length distributions.* We find that the median length of a detected iLOH event in our data is 2.27 kb (IQR = [351, 7018] bp), which is significantly longer than 321 bp (IQR = [105, 2128] bp; $P < 1 \times 10^{-4}$, permutation test) detected with the standard pipeline (Fig. 4c). After correcting for undetected events, the median iLOH event length drops from 2.27 kb to 399 bp (IQR = [102, 4189] bp), which is significantly longer than the corrected median of 102 bp (IQR = [52.5, 229] bp; $P < 0.01$, permutation test) under the standard method. Similarly, we find that the median length of a detected tLOH event in our dataset is 156.4 kb (IQR = [35.0, 424.0] kb), which is a significant increase compared to the median length of 25.5 kb (IQR = [9.8, 230.8] kb; $P < 0.01$, permutation test) detected with the standard approach (Fig. 4c).

### Revised LOH estimates across *S. cerevisiae* hybrids

As mentioned above, previous studies that characterized LOH events in yeast hybrids relied on various versions of the standard single-reference genotyping method. We therefore sought to test the robustness of these previously published estimates of LOH statistics with respect to the genotyping method. To this end, we selected data for 11 intraspecific hybrids of *S. cerevisiae* obtained from mutation accumulation experiments by Pankajam et al. (2020), Sui et al. (2020), and Dutta et al. (2021), and reanalyzed these datasets using our improved genotyping method (see Section "Data" in Materials and Methods for details and Supplementary Data 2 for the list of LOH events detected with our method). A detailed comparison of our revised LOH estimates against the previously published ones is provided in Supplementary Figs. 8 and 9 in Supplementary File 1 and Supplementary Table 3 in Supplementary File 2. We find several differences that are statistically significant and, in some cases, large. Most importantly, for 4 out of 11 hybrids, our revised LOH event rate estimates are significantly higher than previously reported, by between +18% and +426% (factor of ~5.3), and it is lower by between −39 and −81% in 2 hybrids (Wilcoxon rank sum test with Benjamini–Hochberg multiple testing correction and FDR ≤ 0.05; Supplementary Fig. 9 in Supplementary File 1). We did not

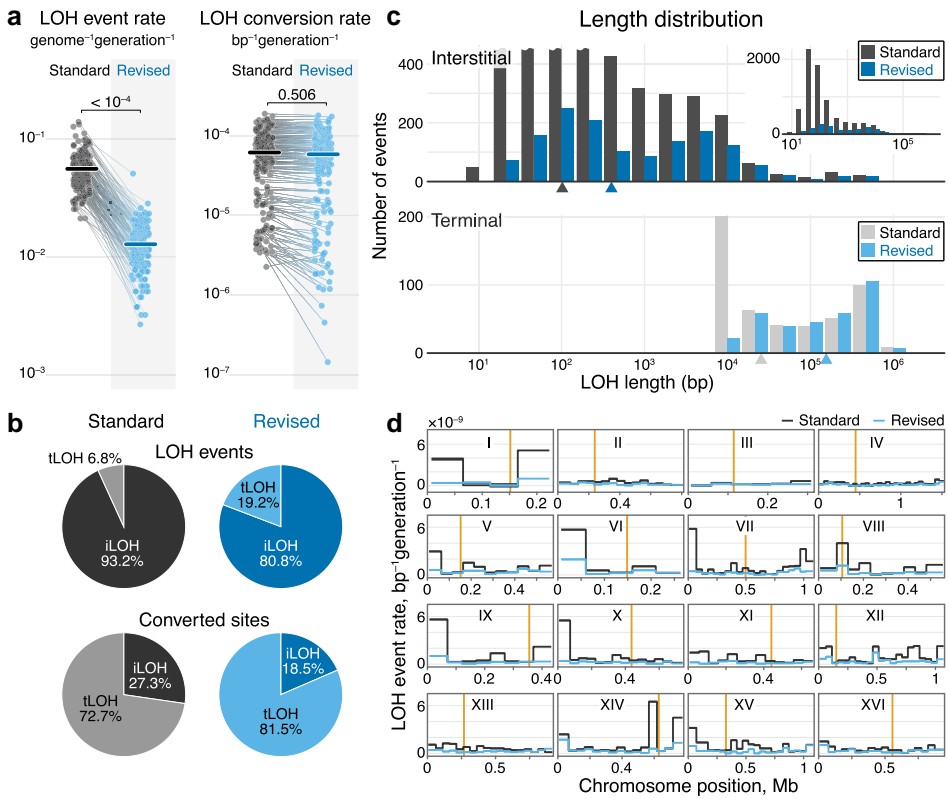

**Fig. 4.** LOH rates and features are significantly different between the standard and revised genotyping methods. a) Differences in LOH event and conversion rates between standard and revised methods. Each end-point clone is represented as a pair of points connected by a line segment. P-values are from a permutation test for the difference of the means. b) The fraction of LOH events and converted sites attributed to interstitial and terminal LOHs under the standard and revised methods. c) Length distributions of iLOH and tLOH events. Event counts in the interstitial plot are truncated at 450 to facilitate comparison and the untruncated distribution is shown in the inset. Triangles point to the distribution median. d) Comparison of local LOH event rates under the standard (black line) and revised (blue line) methods. The DNA breakpoint location was inferred from the boundaries of each LOH event (see Section "Calculation of LOH event and conversion rates" in Materials and Methods). Vertical orange lines indicate centromeres.

find any association between these differences and the characteristics of the hybrids (e.g. heterozygosity, relatedness to S288C).

We next asked whether the corrections resulting from our improved genotype inference procedure alter what we broadly know about the variation in LOH rates and characteristics across yeast hybrids. All quantities we report from this point onwards are based on our new genotyping approach unless otherwise noted.

We find that iLOH event rates vary across hybrids by a factor of about 60, from $2.6 \times 10^{-3}$ to $1.6 \times 10^{-1}$ genome$^{-1}$ generation$^{-1}$ (Fig. 5a), consistent with a previous report (Dutta et al. 2021). In contrast, the variation in tLOH event rates spans only about a 5.6-fold range between $7.4 \times 10^{-4}$ and $4.1 \times 10^{-3}$ genome$^{-1}$ generation$^{-1}$ (Fig. 5b). iLOH and tLOH event rates appear to be uncorrelated across hybrids (Pearson's $\rho = 0.401$, $P = 0.099$; permutation test; Supplementary Fig. 10 in Supplementary File 1), suggesting that variation in these two types of events is driven by different genetic factors. We investigate this hypothesis further in the next section.

Since iLOH events occur more frequently, they comprise a large majority (between 83.5% and 99.2%) of all LOH events in all examined hybrids (Supplementary Fig. 9c in Supplementary File 1). However, since iLOH events are generally much shorter than tLOH events in all hybrids, with median lengths of iLOH and tLOH events ranging from 213 bp to 2.9 kb and 18.41 kb to 156.36 kb, respectively (Fig. 5c and d), both tLOH and iLOH events contribute similarly to the total rates of heterozygosity loss, such that between 11.4% and 82.5% of sites lose heterozygosity due to

tLOH events (Supplementary Fig. 9d in Supplementary File 1). These ranges are broadly consistent with those reported previously (Yim et al. 2014; O'Connell et al. 2015; Dutta et al. 2017, 2021; Loeillet et al. 2020; Pankajam et al. 2020; Sui et al. 2020; Tutaj et al. 2022).

*Homolog biases.* With our improved genotyping method, we find no homolog bias in either iLOH or tLOH events in 5 out of 12 hybrids, including one (S288C × RM) where such bias was measured (though not explicitly discussed) in a previous study (Pankajam et al. 2020). At the same time, we detect significant bias ranging from 51.8% to 65.1% among iLOH events in 6 hybrids and a bias of 76.4% among tLOH events in 1 hybrid (Supplementary Fig. 11 in Supplementary File 1). Previously, only the biases in BTI × ABA a.k.a. H4 and ACD × AKQ a.k.a. H5 were reported (Dutta et al. 2021). We speculate on their possible causes for these biases in the Discussion.

*LOH rates do not correlate with heterozygosity.* As mentioned in the Introduction, Dutta et al. found a positive relationship between the tLOH event rate and hybrid heterozygosity in a set of 9 yeast hybrids (Dutta et al. 2021), which contradicts several other observations (Datta et al. 1997; James et al. 2019; Tattini et al. 2019; Tutaj et al. 2022). To resolve this contradiction, we tested whether the average level of heterozygosity between the hybrid parents correlates with the genome-wide tLOH- and iLOH-event and -conversion rates among our 12 yeast hybrids after the

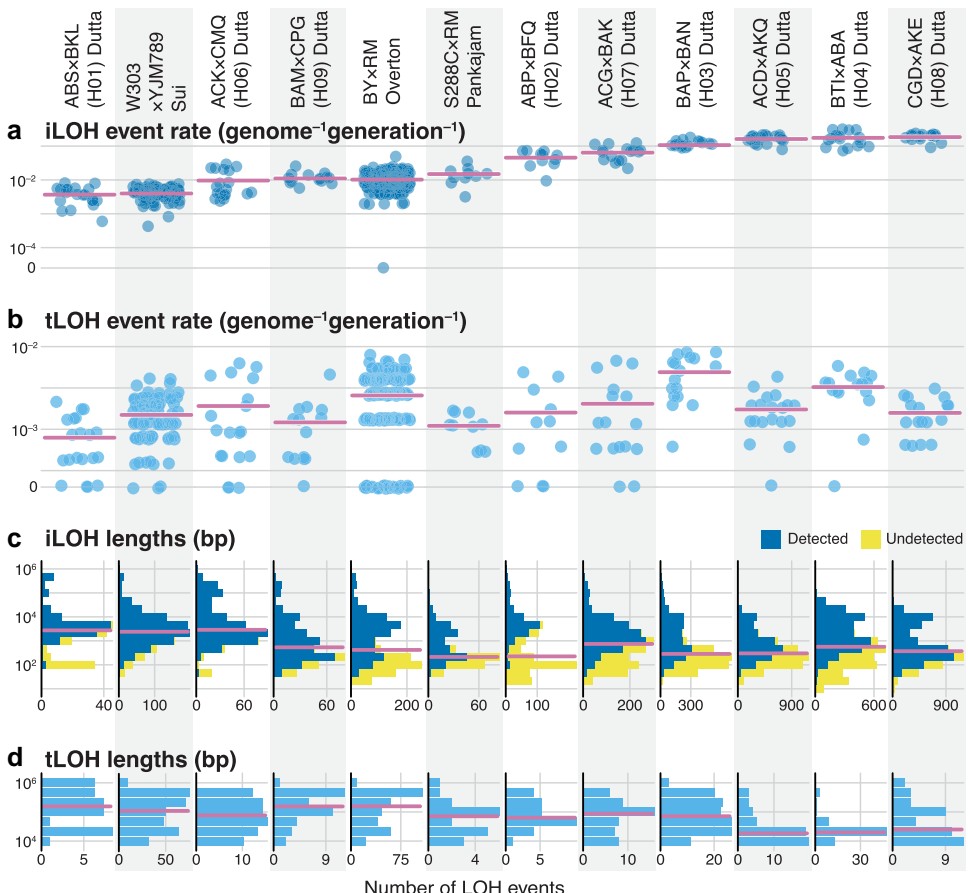

**Fig. 5.** Comparison of LOH features across yeast hybrids. a) iLOH event rates estimated for each end-point clone (blue dots) and mean rates for each hybrid (pink horizontal lines). b) Same as panel a but for tLOH event rates. c) Interstitial LOH event lengths of each hybrid for detected (blue bars) and undetected (yellow bars) events. Pink lines indicate median values. d) Same as panel c, but for tLOH event lengths.

application of our improved genotyping procedure. We find no significant correlation in any of these relationships (Supplementary Fig. 12 in Supplementary File 1), which suggests that the reported association between tLOH event rate and heterozygosity was likely spurious, possibly caused by genotyping errors.

Taken together, the results presented in this section demonstrate that our genotyping method significantly improves the estimates of LOH rates and characteristics. Although this revision does not qualitatively change our understanding of the magnitude and properties of LOH events in yeast, the fact that a previously reported correlation between hybrid heterozygosity and rate of LOH events is not supported by our revised estimates highlights the importance of taking genotyping errors into account.

## Genetic factors affecting the variation in the rates of iLOH and tLOH events

One of the most striking observations, first noted by Dutta et al. (2021), is that the rate of iLOH events varies enormously across hybrids, by about 60-fold according to our revised estimates (Fig. 5a). The rate of tLOH events also varies substantially, by about 6-fold (Fig. 5b), and quite independently of the iLOH rate (Supplementary Fig. 10 in Supplementary File 1). What are the genetic causes of this variation? We can think of this variation as being driven by a mixture of local and genome-wide genetic factors, which we refer to as "cis" and "trans," respectively. An example of a cis factor is the ribosomal RNA gene array located on the right arm of Chr XII, whose size has been shown to affect the rate of tLOH events on that chromosome arm (Sui et al. 2020). Examples of hypothetical

trans-factors are the genome-wide level of heterozygosity or mutations in DNA-repair enzymes (Symington et al. 2014; Coelho et al. 2019; Sui et al. 2020). Both cis- and trans-factors could be either shared between hybrids (i.e. they can be "fixed" in S. *cerevisiae* in the population genetics sense) or they can vary across hybrids (i.e. be "segregating"). Thus, we next sought to assess how much variation in the iLOH and tLOH rates across hybrids and the genome is attributable to fixed or segregating cis factors or segregating trans factors. Note that since the detection of fixed trans effects requires an outgroup, they cannot be detected within this dataset.

To answer this question, we model the per generation rate $\lambda_{ha}$ of LOH events (either interstitial or terminal, separately) on chromosome arm $a$ of hybrid $h$ as

$$\lambda_{ha} = \mu + \alpha_h L_a + \beta_a + \gamma_{ha} \tag{6}$$

where $L_a$ is the length of chromosome arm $a$, and the terms $\mu + \alpha_h L_a$, $\beta_a$ and $\gamma_{ha}$ represent the segregating trans factors, fixed cis factors, and segregating cis factors, respectively. We embed equation (6) into a generalized linear mixed effects modeling (GLMM) framework to model the LOH event counts observed on each chromosome arm of each end-point clone of each hybrid (Supplementary Table 4 in Supplementary File 2) and estimate the variance fractions attributed to each factor as well as to measurement noise (see Section "Analysis of variance in the LOH event rates" in the Materials and Methods for details).

We find that 77.20% of the variance in the iLOH event rates is attributable to segregating trans-factors and 21.47% is noise.

Even though cis factors explain only about 1.32% of variation, each of them is statistically significant ($P = 3.99 \times 10^{-12}$ for fixed cis factors and $P = 6.13 \times 10^{-42}$ for segregating cis factors; Supplementary Table 5 in Supplementary File 2). Consistent with this observation and with previous work (Dutta et al. 2021), we find a strong linear relationship between chromosome-arm length and the rate of iLOH events, with slopes that vary widely across hybrids (Fig. 6a and Supplementary Table 6 in Supplementary File 2).

Since the number of tLOH events in all of the analyzed datasets is much smaller than the number of iLOH events, our power to discriminate between the contributions of different factors to tLOH variation is low. We find that 96.68% of the variation is attributable to noise. The remaining 3.32% are attributable to segregating trans-factors whose contribution is statistically significant ($P = 2.02 \times 10^{-35}$; Supplementary Tables 4 and 5 in Supplementary File 2).

Even though the GLMM approach revealed no significant contribution of cis factors to the variation in tLOH rates, we reasoned that this lack of power could be caused by short chromosomes where we observe very few tLOH events, particularly in hybrids with few MA lines (Fig. 6b; Supplementary Table 4 in Supplementary File 2). In fact, we have a strong prior expectation that cis factors do contribute to tLOH variation across the genome because there is a well-known tLOH event hotspot on Chr. XII, which is associated with the rRNA gene array (see Fig. 6b and Sui et al. 2020). Although this gene array is present in all hybrids, how its length and hence its contribution to tLOH rates vary between them is unknown. In addition, other as yet unknown tLOH hot- or coldspots may exist on other chromosomes (St Charles and Petes 2013; James et al. 2019; Pankajam et al. 2020). Thus, to increase our power to detect such variation, we selected 8 chromosome arms that are longer than 0.5 Mb and asked whether the proportion of tLOH events that occur on each of them differs across hybrids. Indeed, we found significant differences across hybrids on the right arms of Chr. XII and Chr. II ($P = 7.19 \times 10^{-7}$ and $P = 7.54 \times 10^{-3}$, respectively, G-test with 11 degrees of freedom, after Bonferroni correction; Fig. 6b).

In summary, these results show that the variation in iLOH event rates is primarily driven by segregating trans factors, i.e. those that affect the entire genome, whereas the contribution of cis factors to iLOH rates is small. Consequently, the genome-wide distribution of iLOH events is nearly uniform, at least at the chromosome-arm scale, as suggested previously (Sui et al. 2020; Dutta et al. 2021). While we do not have enough statistical power to quantify the relative contributions of different factors to the variation in tLOH rates, we find evidence that these rates are affected by both trans and cis factors that segregate in the yeast population.

## Discussion

The work presented here has both methodological and biological implications, and we discuss these two aspects separately.

### Methodological implications

Detection of LOH events from MA experiments relies on accurate inference of ancestral and descendant genotypes at potentially heterozygous loci. We have shown that the standard single-reference genotyping approach produces a substantial number of both false homozygous and false heterozygous calls (Fig. 2). Both types of errors can result in spurious and often homolog-biased LOH events with a highly nonuniform distribution along the genome (Fig. 1). We developed a dual-reference genotyping method and a statistically grounded heterozygous genotype filter that together drastically reduce the number of dubious calls. We demonstrated that our improved genotyping approach has significantly lower error rates, identifies LOH events with lower homolog biases and results in a smoother genome-wide distribution (Figs. 2–4). We also proposed several simple corrections to account for undetected LOH events.

### The importance of sequencing the founder clones

One important methodological conclusion from our work is that an accurate detection of LOH events in the context of mutation accumulation experiments depends critically on the quality of genotype inference, particularly for ancestral genotypes. Most previous studies have not sequenced the founder clones directly but rather inferred their genotypes from reference sequences. However, the genomes of parental haploids may differ somewhat from these references. Errors in the founder genotypes can lead to detections of correlated false losses of heterozygosity across multiple descendants, which could inflate LOH rates and lead to false inference of homolog biases. Furthermore, sequencing of the founder clones allows one to estimate genotyping error rates and design filters for removing dubious calls, as we have done here. Thus, we recommend that future mutation accumulation studies sequence ancestral clones to high depth, including long-read sequencing, which could resolve possible structural variants (see Section "Recalcitrant HCHB markers can arise from structural variation").

### The problem of quality assessment in genotype calls and LOH inference

One major problem in the detection of LOH events—particularly for the purposes of quantifying their rates and genome-wide distributions—is how to assess the quality of this inference. Short of verifying hundreds of inferred LOH events by low-throughput methods, such as Sanger sequencing, we developed 2 complementary approaches for quality assessment in high-throughput short-read data.

First, we found that plotting the distribution of LOH homolog biases along the genome, as we have done in Fig. 1c, is an effective way to visually assess the quality of LOH inference. The presence of highly converted highly biased (HCHB) regions in the genome suggests problems with genotyping (Minoche et al. 2011; Bobo et al. 2016).

Second, to assess the quality of genotyping more directly, we developed an approach to estimate Hetero- and Homo-FDR, i.e., the expected fractions of sites incorrectly genotyped as heterozygous or homozygous, respectively. Our estimates suggest that the Homo-FDR is on the order of a few percent, which is consistent with a previous estimate (Bobo et al. 2016) and is about 500- to 1,000-fold higher than the Hetero-FDR in our dataset (Fig. 2a). Naively, this Homo-FDR might appear unrealistically high because it seems to imply that ~$10^5$ sites in the genome identified as homozygous are in fact heterozygous. However, this interpretation is incorrect because our FDR estimates pertain not to all sequenced sites (the vast majority of which are monomorphic in all sequenced clones) but only those that are identified by the genotype caller as polymorphic in each family (see Section "Single-reference genotyping" in Materials and Methods). By the design of our experiment, the vast majority of polymorphic sites are heterozygous in all clones, and homozygosity arises occasionally either due to LOH events or genotyping errors. Thus, while our Homo-FDR estimate is high, it is not unreasonable.

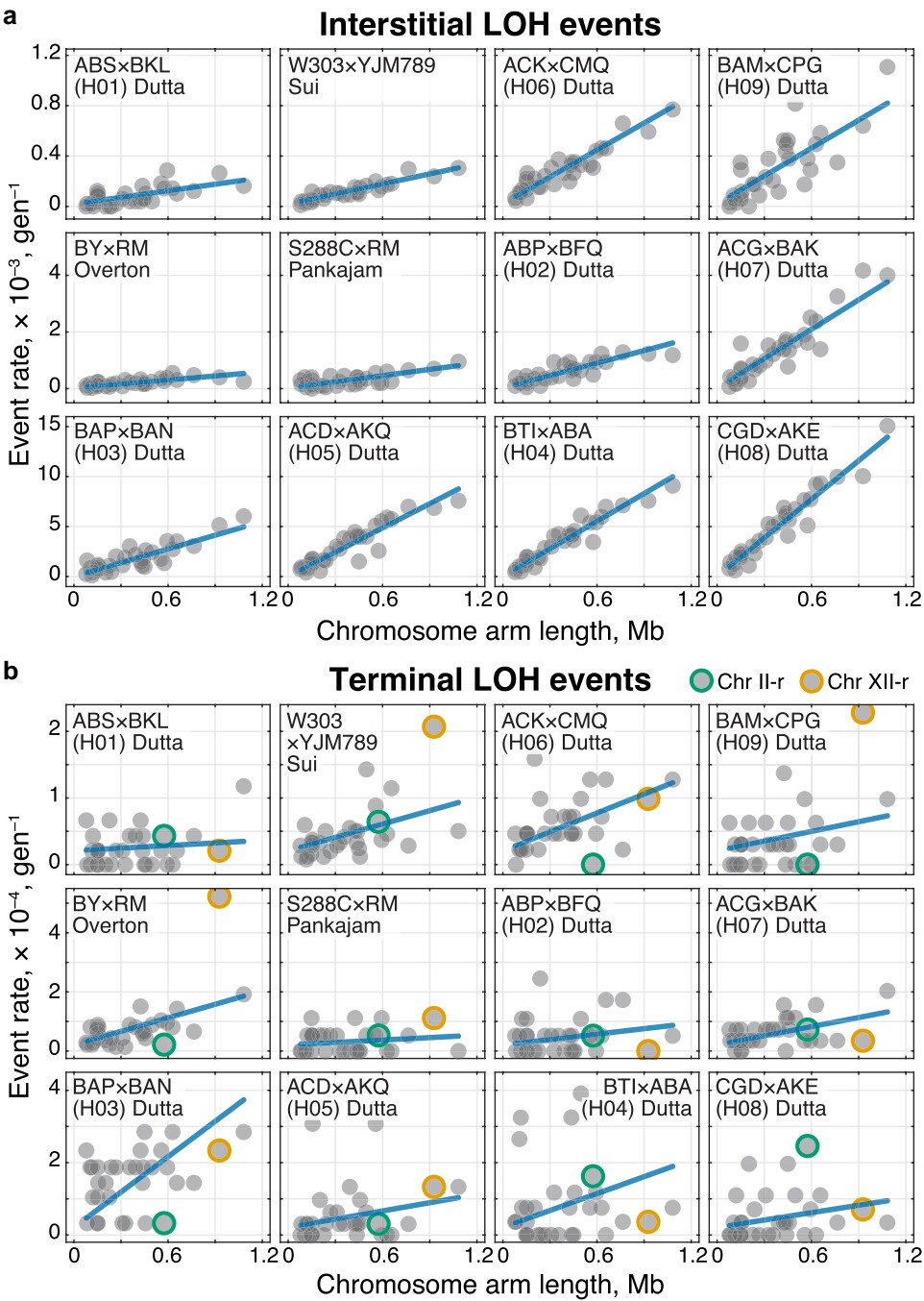

**Fig. 6.** Variation in the iLOH and tLOH event rates across chromosome arms and hybrids. a) iLOH rates for each chromosome arm as a function of its length for each hybrid, as indicated. Blue lines show best-fit GLMM slopes (see Supplementary Table 6 in Supplementary File 2 for best-fit parameter values). b) Same as panel a but for tLOH events. The proportion of tLOH events observed on the right arms of Chr. XII and II are significantly different across hybrids, which is indicated by orange and green outlines, respectively (see text for details). Both iLOH and tLOH rates are calculated based only on detected events, i.e., no corrections for undetected events were applied here.

Since our approach for estimating the Homo- and Hetero-FDR relies on some assumptions, it is possible that our estimates may be wrong if these assumptions are violated. Specifically, we calculate FDR based on a subset of sites in the founders that we deem erroneous based on criteria identified by our model under the assumption that both types of genotyping errors occur with probabilities of at least $10^{-4}$. It is then conceivable that if the error probabilities were in fact much lower than this bound, our criteria may misclassify correctly genotyped sites as erroneous and thereby undermine our FDR estimates. However, our model shows that

our criterion for identifying likely false homozygous sites should remain robust even if the probability of false homozygous errors $\epsilon_{\mathrm{HOM}}$ is much lower than $10^{-4}$, as long as the probability of false heterozygous errors $\epsilon_{\mathrm{HET}}$ also remains sufficiently low (see Supplementary Fig. 2 in Supplementary File 1). Thus, we think that it is unlikely that our high estimate of Homo-FDR is a result of massive misclassification of true homozygous sites as false.

Assuming that our estimated Homo-FDR is correct, it suggests that the quality of homozygous calls at polymorphic sites is substantially lower than the quality of heterozygous calls.

This conclusion is supported by the fact that, at the majority of sites where the genotype calls with respect to two parental genomes disagree, one call is homozygous and the other one is heterozygous, and the homozygous call is usually of lower quality. If a substantial fraction of homozygous genotype calls is indeed problematic, it would be useful to develop an approach for filtering them out. Unfortunately, applying the same filtering approach for homozygous sites as we did for heterozygous sites appears difficult because it is unclear how to identify a set of high-confidence reference homozygous calls among sites that are polymorphic within families.

### Potential causes of genotyping errors

We have investigated the mechanistic causes for the observed genotyping errors in one specific case (left arm of Chr XII) and found that they were caused by structural variation present in the ancestor strains but absent from the reference genome. More broadly, we suspect that genotyping errors can arise because, at least in some genomic regions, the mapping algorithm more often discards or incorrectly maps the nonreference reads, i.e., those that contain polymorphisms relative to the reference. Thus, reads that are successfully mapped are more likely to carry the reference allele. This mechanism can produce a skewed distribution of nonreference read depths observed in Fig. 1b and generate a reference bias in the apparent LOH events.

### Applicability beyond mutation accumulation experiments

Our analyses and methods are directly relevant to mutation accumulation experiments with interspecific hybrids designed for detecting LOH events. However, some of our approaches may be relevant beyond this specific experimental design, e.g. if one is interested in detecting LOH events in wild isolates for which parental genomes required by our dual-reference genotyping pipeline may not be available (Peter et al. 2018). We suspect that the rate of genotyping errors probably depends on the genetic similarity of the sequenced variant to the reference and the bias may be stronger when the reference is more closely related to one homolog than the other. One useful diagnostic tool could be to map sequencing reads against multiple references and compare resulting callsets. A version of our heterozygote filter may also be applicable more broadly if one is able to obtain genotype calls from multiple independent replicates. Finally, our observations should be taken into account when interpreting LOH data. For example, selection that favors a homozygous state at an initially heterozygous locus could result in localized biased LOH events, similar to the spurious HCHB pattern that we saw in Fig. 1c. Thus, such spurious patterns should be ruled out before more biologically interesting explanations, such as selection, can be accepted.

## Biological implications

Using our improved genotyping approach, we revised the estimates of LOH rates and distributions in 12 yeast hybrids. While these revised estimates represent significant improvements for several hybrids, they do not qualitatively change the overall picture of the magnitude of variation in the rates and characteristics of LOH events in yeast, with two exceptions. First, our revised estimates do not support a previously reported positive correlation between hybrid heterozygosity and the rate of tLOH events, at least within the ~10-fold range of heterozygosities sampled here, from 0.083% to 0.97%. (Note that this range includes the roughly 0.2% heterozygosity in human populations (1000 Genomes Project Consortium et al. 2015) but is much less than

in some other fungal species, where heterozygosity can exceed 10% (Baranova et al. 2015)). Second, our estimates of LOH conversion bias differ from those reported in the previous literature.

Finally, we found that the variation in iLOH event rates across yeast hybrids is overwhelmingly driven by trans factors, i.e. those that affect genome-wide iLOH event rates, whereas the variation in tLOH event rates is driven by a combination of both trans and cis factors, the latter being those that affect the rate of tLOH events only in their genomic neighborhood.

### Distinct modes of accumulation of iLOH and tLOH events

Our analyses underscore that iLOH and tLOH events accumulate in genomes of *S. cerevisiae* in markedly different ways. Consistent with previous reports (Sui et al. 2020; Dutta et al. 2021), we found that the rates of iLOH events are highly correlated with chromosome arm length in all hybrids (Fig. 6a). This observation indicates that iLOH events are distributed nearly uniformly across the genome (at least at the chromosome-arm scale, see below) and suggests that the ~60-fold variation in the rates of iLOH events among hybrids is driven primarily by trans factors segregating in the yeast population. In contrast, chromosome arm length is a poor predictor for the rate of tLOH events, though it is statistically significant (Fig. 6b). We found that tLOH events are driven in part by segregating cis factors, i.e. many tLOH events are localized to different chromosome arms in different hybrids. Given these distinct patterns of variation, it is perhaps not surprising that iLOH and tLOH event rates are uncorrelated across hybrids (Supplementary Fig. 10 in Supplementary File 1).

The mode of accumulation of LOH events has implications for how their rates should be reported. Given the near-uniform genomic distribution of iLOH events, it makes sense to report iLOH rates on a per-basepair basis (e.g. as an average value per genome), or at any larger scale of DNA length (e.g. per chromosome arm or per genome). On the other hand, given that the distribution of tLOH events appears fairly nonuniform, a tLOH rate expressed as a genome-wide average per basepair value can be misleading. Instead, tLOH rate is a feature of larger units of DNA, such as chromosome arms or the whole genome. Thus, we suggest reporting tLOH rates on a per chromosome arm or per genome basis.

### Scale of variation of iLOH rates

The fact that our observations are consistent with a fairly uniform distribution of iLOH events across the genome at the chromosome-arm scale does not exclude the possibility that meaningful variation in iLOH event rates exists at smaller scales. For example, Sui et al. found significant associations between iLOH events and replication–termination and high-GC content sequences (Sui et al. 2020). However, the contribution of such small-scale variation to differences in iLOH rates across the entire chromosome arm appears to be very small (we find that less than 2% of variation in iLOH rates is attributed to cis factors). This could be either because these cis factors are rare and/or weak or because their contributions average out on the chromosome-arm scale. Teasing these two possibilities apart will likely require MA experiments with very large numbers of replicate lines. Despite their apparently small contribution to rate variation, cis factors provide the most parsimonious explanation for the observed conversion biases (see below).

### Possible mechanistic causes for the variation in LOH rates across hybrids

While our analysis showed that both cis and trans factors contribute to variation in the LOH rates, what these factors actually are

remains to be determined. Identifying the segregating cis factors that elevate or suppress local rates of tLOH events should be relatively straightforward, at least in yeast, e.g. by using a factorial mating scheme followed by standard MA experiments of appropriate size. Identifying the segregating trans factors may be more difficult because they may be diffuse across the genome, e.g. the overall level of heterozygosity.

Regardless of their actual identity, both cis and trans factors must affect the LOH rates by modulating the usage of different DNA repair pathways. iLOH events can arise via either the synthesis-dependent strand annealing (SDSA) or the double-stranded break repair (DSBR) pathways, while tLOH events can result from DSBR or the break-induced replication (BIR) pathways (Symington et al. 2014; Jinks-Robertson and Petes 2021). If the genetic variation among hybrids primarily affected the DSBR pathway, which contributes to both types of events, we would expect the rates of iLOH and tLOH events to covary across hybrids. Instead, the fact that iLOH and tLOH events exhibit distinct and uncorrelated patterns of variation suggests that the mechanistic causes for the iLOH and the tLOH variation are different. One possibility is that the nearly 60-fold trans-driven variation in iLOH event rates among hybrids is driven by the frequency of engagement of the SDSA mechanism, which does not contribute to tLOH events. The fact that iLOH events tend to be shorter in hybrids with higher iLOH-event rates (Fig. 5c) is consistent with this hypothesis because the SDSA pathway typically produces shorter repair tracks than the DSBR pathway (Guo et al. 2017; Hum and Jinks-Robertson 2017).

Why some repair mechanisms would be used more often in some hybrids than others is another interesting open question. One possibility is that some genetic differences between hybrids affect the regulation of different repair pathways. Another possibility is that hybrids differ in the frequencies of double-stranded DNA breaks that are repaired by certain pathways (Small et al. 2007).

### LOH homolog biases and their possible causes

Several previous studies reported that LOH events in some S. cerevisiae hybrids are biased toward one parental homolog (Loeillet et al. 2020; Pankajam et al. 2020; Dutta et al. 2021). However, these studies used single-reference genotyping methods, which are prone to producing spurious homolog biases, as we have shown here. Our reanalyses of some of these datasets eliminated some of the observed homolog biases, confirmed others, but also uncovered new ones. Overall, we found that LOH events in 7 out of 12 hybrids exhibit statistically significant homolog biases (Supplementary Fig. 11 in Supplementary File 1). While our genotyping method has a significantly reduced rate of genotyping errors, we cannot rule out the possibility that the LOH homolog biases that we detect stem from residual genotyping errors. Alternatively, these residual biases could be biological, in which case it would be interesting to know what could possibly cause them.

Previously, Loeillet et al. (2020) proposed that LOH homolog biases could arise due to lethal allele incompatibilities. In other words, the LOH events themselves may be unbiased, but certain LOH events are not observed because they generate inviable genotypes. While this explanation is plausible in principle, it requires the presence of a rather specific type of epistasis between marker sites whereby both homozygous genotypes AABB and aabb (parental diploids) as well as the double-site heterozygote AaBb (founder diploid) are all viable, while the single-site heterozygote AABb is not. It is unclear whether such epistatic incompatibilities are sufficiently common between closely related yeast strains to explain the LOH biases on the order of a few percent.

We think that LOH homolog biases can be more easily explained by cis-acting factors. Indeed, if the rate of DNA breaks or the type of engaged repair mechanism are at least in part determined by the local DNA-sequence context and if this context is different on the two homologs, we would expect that LOH conversions toward the two homologs would occur at different rates. The well-known rRNA gene array on Chr XII is a good example. Different yeast strains have different numbers of these gene repeats, and each additional repeat elevates the probability of a dsDNA break at this locus (Sui et al. 2020). Thus, the homolog with more repeats is expected to exhibit more dsDNA breaks and therefore more often be converted towards the other homolog. Evaluating all these and possibly other hypotheses that could explain the observed LOH homolog biases is another area for future investigations.

## Data availability

Supplementary Data 1. Sanger sequencing chromatogram files: https://doi.org/10.5281/zenodo.15741043. Supplementary Data 2. LOH event data: https://doi.org/10.5281/zenodo.15750221.

**Code:** https://doi.org/10.5281/zenodo.15748319.

Supplemental material available at GENETICS online.

## Acknowledgments

We thank Kryazhimskiy and Meyer labs for numerous helpful discussions.

## Funding

MSO was supported in part by the Pathways in Biological Sciences Training Program from National Institute of General Medical Sciences program (5T32GM133351). SK acknowledges support from National Institute of General Medical Sciences (R35GM153242).

## Conflicts of interest

The authors declare no competing interests.
See https://doi.org/10.1093/g3journal/jkaf315 in this month's issue of G3: Genes|Genomes|Genetics for a related work.

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

*Editor: D. Barbash*