## [Peer Review File · Genetics]

Improved Genotype Inference Reveals Cis- and Trans-Driven Variation in the Loss-of-Heterozygosity Rates in Yeast

Michael Overton and Sergey Kryazhimskiy

NOTE: The reviews and decision letters are unedited and appear as submitted by the reviewers.

In extremely rare instances and as determined by a Senior Editor or the EIC, portions of a review may be redacted. If a review is signed, the reviewer has agreed to no longer remain anonymous.

The review history appears in chronological order.

Review Timeline:

Submission Date:	2025-09-15
Editorial Decision:	2025-10-23
Revision Received:	2025-12-06
Accepted:	2025-12-11

October 23, 2025

RE: GENETICS-2025-308599

Dear Dr. Kryazhimskiy:

I am pleased to accept your manuscript titled "Improved Genotype Inference Reveals Cis- and Trans-Driven Variation in the Loss-of-Heterozygosity Rates in Yeast" for publication in GENETICS, pending minor revision.

Please submit your revision along with a point-by-point description of how you modified the manuscript in response to the reviewers' concerns and valuable suggestions (which can be viewed at the bottom of this email).

I expect you should be able to submit a revised manuscript within 30 days. A suitably revised manuscript will be acceptable for publication; I don't expect to send it out for review.

Please ensure that you have included a Data Availability Statement at the end of the Materials and Methods section. Details available at <https://academic.oup.com/genetics/content/prep-manuscript>. The DAS should include the accession numbers or DOIs of any data you have placed in public repositories, describe supplemental material, include applicable IRB numbers, and may include specifications for how to properly acknowledge or cite the data.

When revising the ms., please make an effort to shorten it, because that almost always improves a manuscript. We urge authors to heed the advice of Strunk and White: "omit needless words"¹. Follow this link to submit the revised manuscript: Link Not Available

Thank you for submitting this story to Genetics.

Sincerely,

Daniel Barbash
Associate Editor
GENETICS

Approved by:
David Begun
Senior Editor
GENETICS

Reviewer comments:

Reviewer #1 :

Overton and Kryazhimskiy use newly generated data from mutation accumulation lines and re-analyze data from mutation accumulation lines in three previous studies to present methodological advances and new biological insight into loss of heterozygosity mutations in *Saccharomyces cerevisiae*. Here the authors make three important methodological observations and corrections 1) they show that using a single reference genome for mapping reads can result in a reference bias, 2) they show a need for sequencing ancestor/founder and descending clones in mutation accumulation studies, and that there are errors in genotyping that produce false heterozygote and false homozygote calls, and 3) they provide corrections for LOH events that are undetectable due to marker placement. They utilize a dual reference approach and develop procedures for identifying and discarding discordant genotypes. The authors demonstrate that certain intervals of the genome are particularly prone to genotyping errors that produce a reference bias and that upon correction, the distribution of interstitial LOH (iLOH) events is uniform across the genome. This is important for being able to detect true iLOH hotspots and true homolog bias.

The authors then combine data from 4 different mutation accumulation experiments from different labs to provide a re-analyzed and holistic understanding of loss of heterozygosity and its variation in rate across 12 different hybrid strain backgrounds. Conclusions of previous studies of large variation in iLOH rates and more modest but still important variation in terminal LOH rates between different hybrids hold true after corrections are applied, but differ quantitatively. The revised estimates of LOH events per genome are important corrections for our understanding of LOH.

I reviewed this previously at a different journal, and include my revised comments here. This is a well written, thorough, and approachable manuscript that tackles an important mutation type that is particularly prevalent and biologically important in asexual species. While the application of the methods is fairly specific to an experimental design that crosses two strains and that each of these strains has a sequenced genome that can be used to create a reference genome, the approach is straightforward and easy to implement. It will be relevant for mutation accumulation studies and some experimental evolution studies that use hybrids. The correction for homolog bias is particularly important. This is certainly something past and future studies must address, and opens the door for understanding the potential biological bases of such findings.

I have only minor comments:

Can you say anything about why these HCHB intervals seem to have reference mapping biases? And relatedly, can the authors offer any advice on how this might issue might be considered in more general cases when there are not two parent reference genomes (like wild isolates)

P. 12 lines 33-35: While I expected a large drop in iLOH, I was surprised by the decrease in tLOH. Can the authors elaborate on this observation?

Fig 5A - adjust axis labels

P. 16 lines 1-5: Any observations about the genomes, hybrids, etc. regarding when the LOH counts increased or decreased in the revised approach?

Reviewer #3 :

Overton et. al leverages a previously reported mutation accumulation (MA) assay using outcrossed *S. cerevisiae* diploid lines to measure rates of mitotic loss-of-heterozygosity (LOH) in yeast. In doing so, the authors uncover several sources of error unaccounted for or poorly accounted for in previous MA studies focused on LOH rates in yeast. The authors generate and apply methodological and statistical corrections for these errors to both their lines as well as outcrossed diploid yeast strains from previously published MA experiments. Finally, the authors generate a mixed model to explain variation in LOH rates across sequence-divergent yeast lines. There are several main findings here which, as the authors point out in the discussion, can be broken down into methodological and biological:

1) (Methodological) Existing workflows for identifying LOH based on single reference mapping may be inflating estimates of both LOH and homolog bias.

2) (Methodological) Spurious heterozygous calls may be a major source of error that inflates LOH estimates. Statistical filters that remove these as well as a correction for unobserved LOH reduces error associated with determining LOH rates.

3) (Biological) LOH rate, particularly interstitial LOH, may be several fold lower than previous estimates in *S. cerevisiae*.

4) (Biological) There are fewer LOH "hotspots" and a lower rate of homolog bias than previously reported. However, the authors do still report signal for LOH hot spots (mostly attributed to structural variation) and unexplained homolog bias in a reanalysis of previously published MA data.

5) (Biological) There is real variation in LOH rates amongst divergent yeast lines. Much of the variation in iLOH rates is explained by genome-wide traits. However, the LOH variation does not correlate with heterozygosity, which contradicts previous findings.

As the authors point out, rates of mitotic LOH have broad relevance in evolution and cell biology, with particular importance in the evolution of cancer cells. It is also true that past work on LOH in the model *S. cerevisiae* has been contradictory, with studies on the rates and determinants of LOH sometimes conflicting with observations of laboratory evolution outcomes. This paper is a significant contribution to work seeking to understand rates and determinants of LOH. The paper is well-written. The analyses are logical, careful, and well-explained such that they seem simple to reproduce. I believe the main findings here to be strongly supported. The figures are clear and easy to interpret. I have no major concerns.

Minor concerns:

The methodology here applies to mutation accumulation assay strains. Much LOH identification is performed on genomes subject to some form of selection or without selection blunted as strongly as in MA lineages, and/or in assays with lower replication. I think this is obvious, but it might be worth qualifying some statements (such as line 23 in the abstract and the "Methodological implications" section of the discussion) with the limitation of an MA design. Some methods here would be unfeasible outside an MA context, while others might be applied in detection of LOH in non-MA lines. Would the authors have any thoughts on how this study informs detection of LOH in non-MA assays?

There is a good deal of yeast-specific jargon that could be reworded to broaden the biological context of the findings. For example, Line 7 - "mating" can be expanded simply to "sexual reproduction" or "sexual recombination" and "vegetative growth" can be reworded in a way that includes organisms with true somatic tissue, etc.

Related to the above comment, the authors use the word hybrid to describe these heterozygous intraspecific crosses. I realize this is customary in yeast to describe these strain crosses; however, the authors could put a bit of explanation somewhere to underscore that these are not hybrids in the typical sense of the word. This is minor, but the wording may conflate the findings here with work on LOH in true, highly divergent interspecific hybrids.

Line 20. Could these errors also be PCR validated as false heterozygotes for a handful of clones?

Figure 3A. The small inset plot is not explained. I assume it is a zoom-in resolution of the first 60 units of the X-axis?

Line 25. we Sanger sequenced

Line 31 & Figure S12 - From the figure, it appears that the maximum heterozygosity is 1% (120 kB/12 MB). This is higher than average human heterozygosity but lower than other animal systems. It seems as though this finding could be contextualized within the range of observed heterozygosity.

Methods

Maybe I'm missing it, but it's not clear which reference genome was used to produce the RM portion of figure 1A&C. Is it the RM assembly in tables S10?

Reference genomes for dual-reference genotyping were modified to match parental haploid strains. This was not done for single-reference genotyping. From the text in the results, it seemed as though the exact same reference genomes were used for single and dual reference genotyping. Do these polished references decrease rates of homolog bias in single-reference genotyping?

Associate Editor comments:

Reviewer #1

Overton and Kryazhimskiy use newly generated data from mutation accumulation lines and re-analyze data from mutation accumulation lines in three previous studies to present methodological advances and new biological insight into loss of heterozygosity mutations in *Saccharomyces cerevisiae*. Here the authors make three important methodological observations and corrections 1) they show that using a single reference genome for mapping reads can result in a reference bias, 2) they show a need for sequencing ancestor/founder and descending clones in mutation accumulation studies, and that there are errors in genotyping that produce false heterozygote and false homozygote calls, and 3) they provide corrections for LOH events that are undetectable due to marker placement. They utilize a dual reference approach and develop procedures for identifying and discarding discordant genotypes. The authors demonstrate that certain intervals of the genome are particularly prone to genotyping errors that produce a reference bias and that upon correction, the distribution of interstitial LOH (iLOH) events is uniform across the genome. This is important for being able to detect true iLOH hotspots and true homolog bias.

The authors then combine data from 4 different mutation accumulation experiments from different labs to provide a re-analyzed and holistic understanding of loss of heterozygosity and its variation in rate across 12 different hybrid strain backgrounds. Conclusions of previous studies of large variation in iLOH rates and more modest but still important variation in terminal LOH rates between different hybrids hold true after corrections are applied, but differ quantitatively. The revised estimates of LOH events per genome are important corrections for our understanding of LOH. I reviewed this previously at a different journal, and include my revised comments here. This is a well written, thorough, and approachable manuscript that tackles an important mutation type that is particularly prevalent and biologically important in asexual species. While the application of the methods is fairly specific to an experimental design that crosses two strains and that each of these strains has a sequenced genome that can be used to create a reference genome, the approach is straightforward and easy to implement. It will be relevant for mutation accumulation studies and some experimental evolution studies that use hybrids. The correction for homolog bias is particularly important. This is certainly something past and future studies must address, and opens the door for understanding the potential biological bases of such findings.

Author response. We thank the reviewer for this positive assessment of our work.

I have only minor comments:

Can you say anything about why these HCHB intervals seem to have reference mapping biases? And relatedly, can the authors offer any advice on how this might issue might be considered in more general cases when there are not two parent reference genomes (like wild isolates)

Author response. We think that the mapping algorithm is more likely to discard or incorrectly map the non-reference reads, i.e., those that contain polymorphisms relative to the reference. Thus, the remaining correctly mapped reads are more likely to carry the reference allele. We have added a short comment on this to the Discussion (P 17 L 37 – P 18 L 6).

Regarding the second question, we think that the read mapping bias is likely always present to some degree, but how it would affect the results probably depends on the research question. As we have shown, estimates of LOH rates are quite sensitive to genotyping errors but other applications (e.g., estimates of heterozygosity or Fst) may be more robust. One obvious approach that might to some extent help correct the mapping bias is to use multiple reference genomes. The reference genomes don't necessarily have to be direct ancestors (though this is likely the best option when available), but mapping against two or more reference genomes might reveal at least some regions where mapping is unreliable. A version of our heterozygote filter may also be applicable more generally. We added these speculative ideas to the Discussion (P 18 LL 7–21).

P. 12 lines 33-35: While I expected a large drop in iLOH, I was surprised by the decrease in tLOH. Can the authors elaborate on this observation?

Author response. The reduction in tLOH events comes almost exclusively from events that are supported by a single marker at the end of a chromosome (see Figure 4C bottom; compare the first light gray bar and the first light blue bar). We have added a comment on this on P 12 L 3–4.

Fig 5A - adjust axis labels

Author response. We could not figure out what the reviewer meant specifically, so we left the figure as is.

P. 16 lines 1-5: Any observations about the genomes, hybrids, etc. regarding when the LOH counts increased or decreased in the revised approach?

Author response. We investigated whether there were any associations between hybrid identity and the effect of our revised approach on the LOH rate estimates, but found none. We now state this explicitly on P 13 LL 15–17.

Reviewer #3

Overton et. al leverages a previously reported mutation accumulation (MA) assay using outcrossed *S. cerevisiae* diploid lines to measure rates of mitotic loss-of-heterozygosity (LOH) in yeast. In doing so, the authors uncover several sources of error unaccounted for or poorly accounted for in previous MA studies focused on LOH rates in yeast. The authors generate and apply methodological and statistical corrections for these errors to both their lines as well as outcrossed diploid yeast strains from previously published MA experiments. Finally, the authors generate a mixed model to explain variation in LOH rates across sequence-divergent yeast lines. There are several main findings here which, as the authors point out in the discussion, can be broken down into methodological and biological:

- 1) (Methodological) Existing workflows for identifying LOH based on single reference mapping may be inflating estimates of both LOH and homolog bias.
- 2) (Methodological) Spurious heterozygous calls may be a major source of error that inflates LOH estimates. Statistical filters that remove these as well as a correction for unobserved LOH reduces error associated with determining LOH rates.
- 3) (Biological) LOH rate, particularly interstitial LOH, may be several fold lower than previous estimates in *S. cerevisiae*.
- 4) (Biological) There are fewer LOH "hotspots" and a lower rate of homolog bias than previously reported. However, the authors do still report signal for LOH hot spots (mostly attributed to structural variation) and unexplained homolog bias in a reanalysis of previously published MA data.
- 5) (Biological) There is real variation in LOH rates amongst divergent yeast lines. Much of the variation in iLOH rates is explained by genome-wide traits. However, the LOH variation does not correlate with heterozygosity, which contradicts previous findings.

As the authors point out, rates of mitotic LOH have broad relevance in evolution and cell biology, with particular importance in the evolution of cancer cells. It is also true that past work on LOH in the model *S. cerevisiae* has been contradictory, with studies on the rates and determinants of LOH sometimes conflicting with observations of laboratory evolution outcomes. This paper is a significant contribution to work seeking to understand rates and determinants of LOH. The paper is well-written. The analyses are logical, careful, and well-explained such that they seem simple to reproduce. I believe the main findings here to be strongly supported. The figures are clear and easy to interpret. I have no major concerns.

Author response. We are pleased with the reviewer's positive assessment of our work.

Minor concerns:

The methodology here applies to mutation accumulation assay strains. Much LOH identification is performed on genomes subject to some form of selection or without selection blunted as strongly as in MA lineages, and/or in assays with lower replication. I

think this is obvious, but it might be worth qualifying some statements (such as line 23 in the abstract and the "Methodological implications" section of the discussion) with the limitation of an MA design. Some methods here would be unfeasible outside an MA context, while others might be applied in detection of LOH in non-MA lines. Would the authors have any thoughts on how this study informs detection of LOH in non-MA assays?

Author response. We agree that our method is limited to the MA design, and we now qualify this in the abstract and several places throughout the text. We also added some thoughts on what aspects of our work might have implications beyond the MA setup (P 18 LL 7–21).

There is a good deal of yeast-specific jargon that could be reworded to broaden the biological context of the findings. For example, Line 7 - "mating" can be expanded simply to "sexual reproduction" or "sexual recombination" and "vegetative growth" can be reworded in a way that includes organisms with true somatic tissue, etc.

Author response. We agree that the language used throughout the paper should be as accessible as possible. We have gone through the text and changed various yeast-specific terms to more general ones where applicable.

Related to the above comment, the authors use the word hybrid to describe these heterozygous intraspecific crosses. I realize this is customary in yeast to describe these strain crosses; however, the authors could put a bit of explanation somewhere to underscore that these are not hybrids in the typical sense of the word. This is minor, but the wording may conflate the findings here with work on LOH in true, highly divergent interspecific hybrids.

Author response. Thank you for this. We now clarify in several places in the text that our hybrids are intraspecific and provide the relevant ranges of heterozygosity on P 18 LL 28–32.

Line 20. Could these errors also be PCR validated as false heterozygotes for a handful of clones?

Author response. In principle, we could confirm the genotypes (e.g. by Sanger sequencing) for some set of clones at those sites rejected by our filter. However, we are not sure what this would accomplish. Our goal here is to remove sites that tend to produce unreliable genotype calls. Some (perhaps many) of these sites may in fact be a heterozygous, but if they cannot be reliably genotyped, we still wish to exclude them.

Figure 3A. The small inset plot is not explained. I assume it is a zoom-in resolution of the first 60 units of the X-axis?

Author response. Correct. We added an explanation to the figure caption.

Line 25. we Sanger sequenced

Author response. Corrected.

Line 31 & Figure S12 - From the figure, it appears that the maximum heterozygosity is 1% (120 kB/12 MB). This is higher than average human heterozygosity but lower than other animal systems. It seems as though this finding could be contextualized within the range of observed heterozygosity.

Author response. Good idea. We now include a passage to provide this context (P 18 LL 28–32).

Maybe I'm missing it, but it's not clear which reference genome was used to produce the RM portion of figure 1A&C. Is it the RM assembly in tables S10?

Author response. Thank you for pointing out this omission. We added a clarification in the figure caption to indicate the source of the RM reference genome.

Reference genomes for dual-reference genotyping were modified to match parental haploid strains. This was not done for single-reference genotyping. From the text in the results, it seemed as though the exact same reference genomes were used for single and dual reference genotyping. Do these polished references decrease rates of homolog bias in single-reference genotyping?

Author response. This is correct, but the results do not change when we modify the reference strains to match the parents in single-reference genotyping. We added a clarification to this effect (P 6 LL 16–19).

December 11, 2025

RE: GENETICS-2025-308599R1

Dr. Sergey Kryazhimskiy
University of California San Diego
Ecology, Behavior and Evolution
9500 Gilman Dr
La Jolla, California 92093

Dear Dr. Kryazhimskiy,

Congratulations, your manuscript titled "Improved Genotype Inference Reveals Cis- and Trans-Driven Variation in the Loss-of-Heterozygosity Rates in Yeast" is accepted for publication in GENETICS! Many thanks for submitting your research to the journal.

To Proceed to Publication:

1. Format your article according to GENETICS style: <https://academic.oup.com/genetics/pages/author-guidelines>
2. Ensure that you comply with data and community resource citation guidelines: <https://academic.oup.com/genetics/pages/author-guidelines#section-5-9-2>
3. Upload your final files at <https://genetics.msubmit.net>
4. Add oupsupport@scipris.com and genetics.oup@novatechset.com (or the domains @scipris.com and @novatechset.com) to your email program's "safe senders" list. You will be contacted by both at various points during the production process.

Notes:

- Your currently-accepted manuscript (unedited, as submitted, reviewed, and accepted) will be published at GENETICS and deposited into PubMed as an Advance Access article. Notify sourcefiles@thegsajournals.org before signing your license if you do not wish to publish your article via Advance Access.
- We invite you to submit an original color figure related to your paper for consideration as cover art. Please email your submission to the editorial office or upload it with your final files. You can submit a small-sized image for evaluation, and if selected, the final image must be a TIFF file 2513px wide by 3263px high (8.375 by 10.875 inches; resolution of 600ppi). Please avoid graphs and small type.
- After files are sent to Oxford University Press we use SciPris to manage article licensing and payment. If you do not have a SciPris account, you will receive an email from no-reply@scipris.com to sign up to use Oxford University Press' author portal. After logging in, follow the online instructions to sign your license and arrange any payment due.

If you have any questions or encounter any problems while uploading your accepted manuscript files, please email the editorial office at sourcefiles@thegsajournals.org.

Sincerely,

Daniel Barbash
Associate Editor
GENETICS

Approved by:
David Begun
Senior Editor
GENETICS

Review comments (if applicable):